# Antibody binding reports spatial heterogeneities in cell membrane organization

Daniel P. Arnold [1], Yaxin Xu [1] & Sho C. Takatori [1] ✉

The spatial organization of cell membrane glycoproteins and glycolipids is critical for mediating the binding of ligands, receptors, and macromolecules on the plasma membrane. However, we currently do not have the methods to quantify the spatial heterogeneities of macromolecular crowding on live cell surfaces. In this work, we combine experiment and simulation to report crowding heterogeneities on reconstituted membranes and live cell membranes with nanometer spatial resolution. By quantifying the effective binding affinity of IgG monoclonal antibodies to engineered antigen sensors, we discover sharp gradients in crowding within a few nanometers of the crowded membrane surface. Our measurements on human cancer cells support the hypothesis that raft-like membrane domains exclude bulky membrane proteins and glycoproteins. Our facile and high-throughput method to quantify spatial crowding heterogeneities on live cell membranes may facilitate monoclonal antibody design and provide a mechanistic understanding of plasma membrane biophysical organization.

Physical crowding of the cell surface glycocalyx has been shown recently to alter the biophysical properties of membranes in a manner that significantly impacts cell function. These alterations include membrane bending, stretching, and fission on reconstituted lipid bilayers[1–5], as well as tubulation and fission in the plasma membranes of cultured cells[6,7]. In addition to inducing membrane deformation, cell surface crowding also modulates the physical accessibility of surface receptors to large soluble ligands and macromolecules[8]. Experiments on reconstituted membranes with grafted synthetic polymers or purified proteins further confirm a decrease in protein binding affinity with increasing grafting density[9–12]. Most clinical monoclonal antibody drugs that rely on direct effector-cell activity are known to target antigen receptors that are buried deep inside the glycocalyx, often within 10 nm from the membrane surface[13,14], suggesting that their effectiveness may be highly dependent upon crowding near the receptor. However, there are currently no methods to characterize the piconewton-scale forces generated by the crowding of ~10 nm cell surface proteins[15].

In addition to surface-orthogonal variations, the mammalian plasma membrane composition is also laterally-heterogeneous, with nanometer-scale protein and lipid clusters forming and dissipating on sub-second timescales[15,16]. In giant plasma membrane vesicles (GPMVs) isolated from cells, Gurdap et al. showed that liquid-ordered membrane microdomains exclude proteins with bulky, heavily glycosylated extracellular domains[17]. Given the lateral and membrane-orthogonal heterogeneity of the glycocalyx, a complete picture of the crowding profile requires three-dimensional (3D) characterization.

While techniques like electron microscopy enable nanometer-scale characterization of the plasma membrane, the preparation process is destructive[18], leaving a need for appropriate molecular probes to study these complex, dynamic systems in vivo[15]. Recently, Houser et al.[19] quantified the surface pressures on reconstituted crowded membranes by measuring the separation distance between FRET fluorophores that stretch due to steric interactions within the brush. The stretching distance in polymer brushes depends weakly on surface density, as height scales with chain density according to $h \sim n^{1/3}$ in the brush regime[20,21] and follows even weaker scaling in the mushroom regime[22]. Therefore, the technique may lose accuracy at the crowding densities observed in physiological surface densities on live cells.

[1]Department of Chemical Engineering, University of California, Santa Barbara, Santa Barbara, CA 93106, USA. ✉e-mail: stakatori@ucsb.edu

In this work, we develop synthetic antigen sensors with precise spatial localization and measure the binding affinity of complementary immunoglobulin G (IgG) monoclonal antibodies in these local crowding environments. We leverage a technique developed recently by Takatori and Son et al.[23], in which a macromolecular probe is introduced to the extracellular side of a plasma membrane to quantify the local osmotic pressure posed by the crowded cell surface via a reduction in effective binding affinity. We advance this technique by enabling spatial localization of the binding site to measure the membrane-orthogonal crowding heterogeneity on both reconstituted membranes and red blood cell (RBC) surfaces. We then reconstruct these systems in-silico, combining proteomics with molecular dynamics (MD) simulations and experiments to map RBC glycocalyx crowding with nanometer-scale spatial precision. Using targeted antigen probes, we expand our spatial resolution laterally, in the plane of the membrane, measuring differences in crowding between plasma membrane domains on live tumor cells. Our findings support the hypothesis that raft-like domains of native membranes exclude proteins with bulky extracellular domains, consistent with the findings of Gurdap et al.[17] on GPMVs. Our simple IgG binding assay to probe spatial heterogeneities on native cell membranes suggests an important role of structural complexities in glycocalyx organization.

## Results

### Synthetic antigen sensors report crowding heterogeneities with nanometer height resolution

The glycocalyx is heterogeneous in both composition and density, which vary as a function of distance from the membrane surface (henceforth "height"). Height heterogeneities in crowding can arise from variations in protein sizes[24–27] and also from polymer brush dynamics of disordered glycoproteins like mucins in the glycocalyx[6,8,28–30]. To characterize the cell surface height heterogeneity, we developed a noninvasive synthetic antigen sensor that inserts into the lipid membrane using a cholesterol tag conjugated to a polyethylene glycol (PEG) linker and a fluorescein isothiocyanate (FITC) fluorophore (Fig. 1A). We developed a family of cholesterol-PEG-FITC sensors with varying PEG linker lengths to adjust the height of the FITC antigen presented above the membrane. After presenting the antigen sensors on the cell surface, we obtain the effective binding avidity of anti-FITC ($\alpha$ FITC) IgG antibody as a function of antigen height.

The PEG linker enables the FITC antigen to sample a distribution of heights above the membrane, while the mean height, $\langle h \rangle$, increases with the molecular weight of PEG. We used cell surface optical profilometry (CSOP)[31] to measure $\langle h \rangle$ of the FITC antigen for sensor linker lengths of 0.5 kDa PEG (PEG0.5k), 2k, 5k, and 10k using silica beads coated with a 1,2-dioleoyl-sn-glycero-3-phosphocholine (DOPC) supported lipid bilayer (SLB). We recovered the predicted increase in $\langle h \rangle$ with molecular weight (Fig. 1B), suggesting that the antigen is probing different crowding microenvironments as a function of linker length.

To validate that our different sensors are probing the height heterogeneities of a crowded membrane surface, we measured $\alpha$ FITC IgG binding to our antigen sensors on a reconstituted glycocalyx-mimetic PEG brush. Our reconstituted SLB on 4 $\mu$m silica beads included 3% 1,2-dioleoyl-sn-glycero-3-phosphoethanolamine-N-[methoxy(polyethylene glycol)-2000] (DOPE-PEG2k) to act as a repulsive brush, with synthetic antigen sensors of a single type inserted into the outer membrane leaflet (see Materials and Methods). Beads were incubated in varying concentrations of fluorescently-labeled $\alpha$ FITC antibodies and allowed to reach equilibrium before fluorescence intensities of beads were collected via fluorescence microscopy. Intensities were fit to a Hill binding isotherm to calculate the dissociation constant $K_D$ (see Supplementary Note 2, Supplementary Fig. S3). The ratio of $K_D$ on the PEG-crowded SLB to that on a bare SLB with no PEG crowders, $K_D/K_{D,0}$, decreases toward unity as the average FITC height increases (Fig. 1C). The FITC antigen on our 10k antigen sensor samples the majority of its height distribution above the DOPE-PEG2k steric brush and has an $\alpha$ FITC binding avidity that is essentially unchanged from the bare membrane value. In contrast, the 0.5k antigen sensor is buried deep inside the PEG brush and the accessibility of the FITC antigen is hindered by a factor of six (Fig. 1C). Our results are consistent with classical polymer brush theory, which predicts a monotonic decrease in brush monomer density with height[28] and a reduction in the effective adsorption energy of a globular protein onto a brush-coated surface[32].

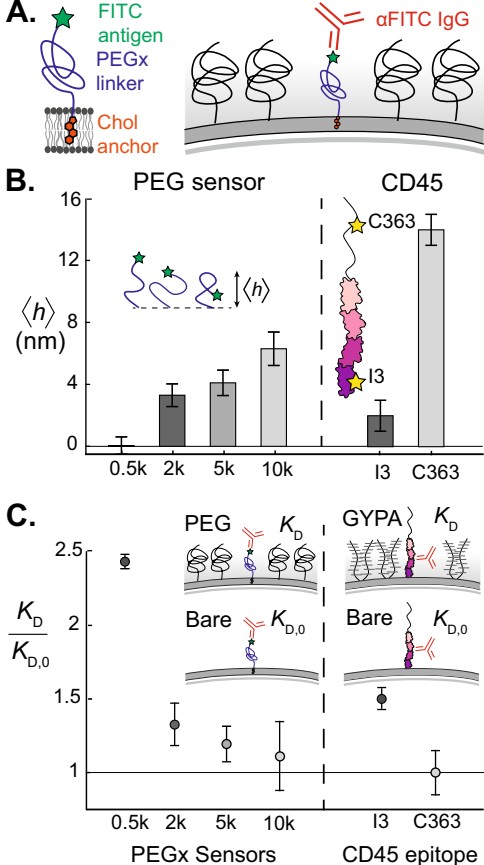

**Fig. 1 | Synthetic antigen sensors enable precise localization and measurement of IgG binding avidity on crowded membrane surfaces. A** Cholesterol-PEG-FITC sensors insert into the membrane and present the FITC antigen at varying heights above the membrane, depending on the PEG linker length. Sensors are exogenously inserted into reconstituted or live plasma membranes, and $\alpha$ FITC IgG avidity is a direct reporter of local crowding. **B** (Left) Mean height $\langle h \rangle$ of synthetic antigen sensors increases as a function of PEG linker molecular weight, as measured by CSOP[31]. Error bars represent standard error of the mean for $n = 65$, 81, 79, and 54 beads, respectively. (Right) Epitope heights of two different $\alpha$ CD45 antibodies, $\alpha$ I3 and $\alpha$ C363 on the transmembrane tyrosine phosphatase CD45, as reported by Son et al.[31]. Error bars correspond to -±1 nm error, reported by Son et al. **C** (Left) Dissociation constants of $\alpha$ FITC to the synthetic antigen sensors on a supported lipid bilayer, containing 3% DOPE-PEG2k as a surface crowder. Dissociation constants are normalized by the bare membrane value, $K_{D,0}$. (Right) Two $\alpha$ CD45 antibodies with distinct epitope heights experience a significant difference in normalized avidity on a reconstituted membrane crowded with a mucin-like glycoprotein, Glycophorin A (GYPA). Points are derived from binding isotherm curve fits, and error bars are derived by fitting curves to data points 1 standard error of the mean above and below each data point, propagated with the 95% CI error of the fits. Curve fits for $K_{D,0}$ are generated from populations of $n = 912$, 905, 1062, 635, 1730, and 1787 beads, respectively. Curve fits for $K_D$ are generated from populations of $n = 2115$, 1281, 1237, 578, 2180, and 3333 beads, respectively.

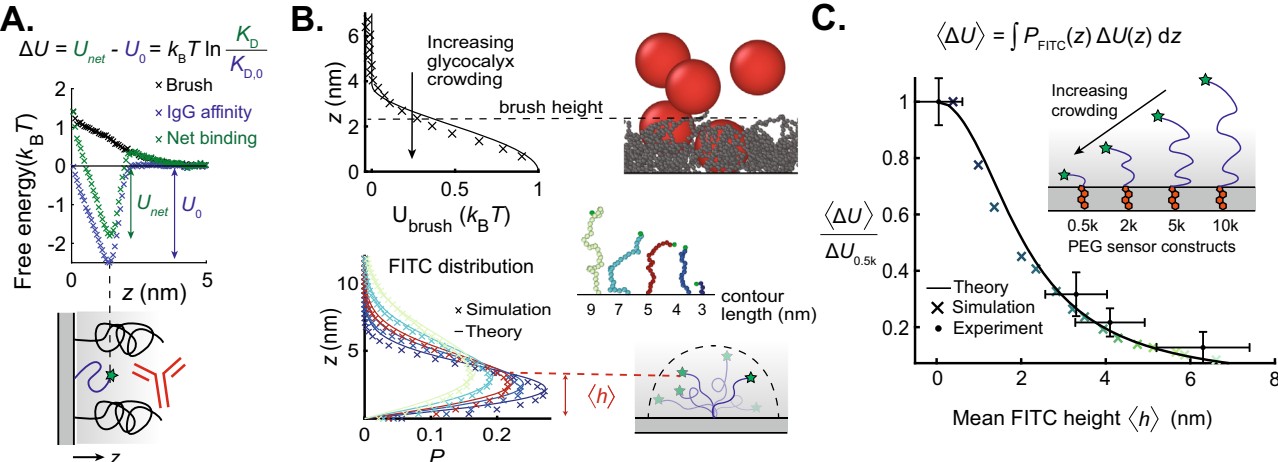

**Fig. 2 | The binding avidity of macromolecules on crowded surfaces is a direct reporter of the energetic penalty posed by the crowded surface. A** Coarse-grained molecular dynamics (MD) simulations of the IgG free energy versus height above the surface. The observed IgG avidity ($U_{net}$, green) is a superposition of an attractive enthalpic binding energy ($U_0$, blue) and a crowding-induced penalty due to crowding polymers ($\Delta U$, black). Therefore, the normalized dissociation constants from Fig. 1C report the local energy penalty of the crowded surface. **B** (Top left) MD simulations of crowding penalty are re-plotted from Fig. 2A, with solid line indicating classical polymer brush theory[28]. (Top right) Simulation snapshot of antibody (red) on polymer (gray) coated surface. Weighting the crowding penalty by the FITC distribution yields a mean crowding penalty $\langle \Delta U \rangle$ for a sensor of given $\langle h \rangle$. (Bottom left) FITC position probability distributions for different polymer contour lengths based on continuous Gaussian chain model[34] (curve) and MD simulations (crosses). (Bottom right) MD snapshots of sensors of different linker lengths and illustration of average height $\langle h \rangle$ of a fluctuating FITC antigen. **C** Antibody avidity data from Fig. 1C are re-plotted to report steric crowding energy as a function of the mean FITC antigen height from Fig. 1B (black circles). Energies are normalized by the smallest antigen sensor size, $\Delta U_{0.5k} = 0.9 k_B T$. Theoretical prediction based on polymer brush theory (curve) agrees with experimental data and MD simulations (crosses). Horizontal error bars are generated based on CSOP data presented in Fig. 1, using populations of $n = 65, 81, 79$, and 54 beads. Experimental data points are derived from binding isotherm curve fits, and vertical error bars are derived by fitting curves to data points 1 standard error of the mean above and below each data point, propagated with the 95% CI error of the fits. Curve fits for $K_{D,0}$ are generated from populations of $n = 912, 905, 1062$, and 635 beads, respectively. Curve fits for $K_D$ are generated from populations of $n = 2115, 1281, 1237$, and 578 beads, respectively.

Based on our results for synthetic sensors on a PEG brush surface, we hypothesized that the height-dependent avidity of IgG would also apply to protein antigens buried within a crowded surface of other membrane proteins. To investigate, we reconstituted an SLB containing 5% 1,2-dioleoyl-sn-glycero-3-[(N-(5-amino-1-carboxypentyl)imino-diacetic acid)succinyl] (DGS-NTA) and created a crowded surface of poly-histidine tagged glycoprotein, Glycophorin A (GYPA). Instead of synthetic antigen sensors, we tethered a dilute surface density of tyrosine phosphatase CD45 on the SLB among the crowded excess of GYPA. As a readout of GYPA crowding, we used $\alpha$CD45 antibodies that target two different epitope sites: pan-CD45 I3 epitope on the first FN3 domain ($\langle h \rangle = 2.5$ nm), and $R_B$ isoform epitope C363 on the upper mucin-like domain ($\langle h \rangle = 15$ nm).

Using CSOP[31], we measured the height of an $\alpha$GYPA monoclonal antibody (clone HIR2) that binds to the N-terminus of GYPA, and found that the average GYPA height is $\approx 12$ nm tall. Thus we expected the C363 epitope on CD45 to explore uncrowded regions above the GYPA brush, while the I3 epitope to remain buried within the brush. Indeed, the relative avidities of $\alpha$C363 and $\alpha$I3 agree with this hypothesis, as $K_D/K_{D,0}$ is $\approx 1$ for $\alpha$C363 while it is $\approx 1.5$ for $\alpha$I3 (Fig. 1C). The consistent correlation between increasing antigen height, $\langle h \rangle$, and decreasing dissociation constant, $K_D/K_{D,0}$, on both PEG and protein brushes confirms that antibody avidity is a robust metric of local crowding.

## Macromolecular binding is a direct reporter of steric energies on crowded surfaces

In this section, we aim to obtain a direct relation between the antibody binding avidity and the local steric free energy penalty of a crowded surface. We combine polymer brush theories with coarse-grained MD simulations to obtain a mechanistic understanding of our synthetic antigen sensors and their applicability on crowded membrane surfaces.

To characterize the energy profile on the membrane, we separately simulated free antibody insertion into a surface-tethered PEG2k brush and antibody binding to surface-tethered sensors to obtain the repulsive penalty associated with crowding, $\Delta U$, (Fig. 2A), and the attractive binding free energy, $U_0$, respectively (see Materials and Methods). We invoke the theory of Halperin[32] and hypothesize that the effective antibody binding free energy on a crowded interface, $U_{net}$, is a superposition of $U_0$ and $\Delta U$. The bare membrane-binding avidity reports the attractive enthalpic term $U_0 = k_B T \ln K_{D,0}$, so that the repulsive entropic energy penalty posed by the brush is given by

$$\Delta U = U_{net} - U_0 = k_B T \ln \left( \frac{K_D}{K_{D,0}} \right). \quad (1)$$

Antibody insertion into the brush reduces the volume available to the polymer and is entropically disfavored. This repulsive energy barrier, $\Delta U$, is proportional to the osmotic pressure, $\Pi$, which scales with monomer volume fraction $\phi$ as $\Pi \sim \phi^{9/4}$[22,32]. The Milner, Witten, and Cates[28] self-consistent field description of a polymer brush predicts a parabolic monomer distribution, so the crowding penalty $\Delta U$ follows a stretched parabolic profile (Fig. 2B, see Supplementary Note 1 for analytical form). Kenworthy et al. showed experimentally that the pressure between apposite membrane-tethered PEG brushes under compression varies with distance according to a profile derived from Milner theory[33]. We, therefore, invoke this theory to describe the form of our PEG2k crowding penalty, which we verify using MD simulations (see Materials and Methods, Supplementary Movie 2).

The flexibility of the PEG linker in our synthetic antigen sensors causes the antibody to bind across a distribution of FITC heights for any given sensor. Thus, we define our experimentally-measured crowding free energy for a given sensor as a mean energy penalty $\langle \Delta U \rangle$, which can be predicted by weighting the FITC 1-D probability

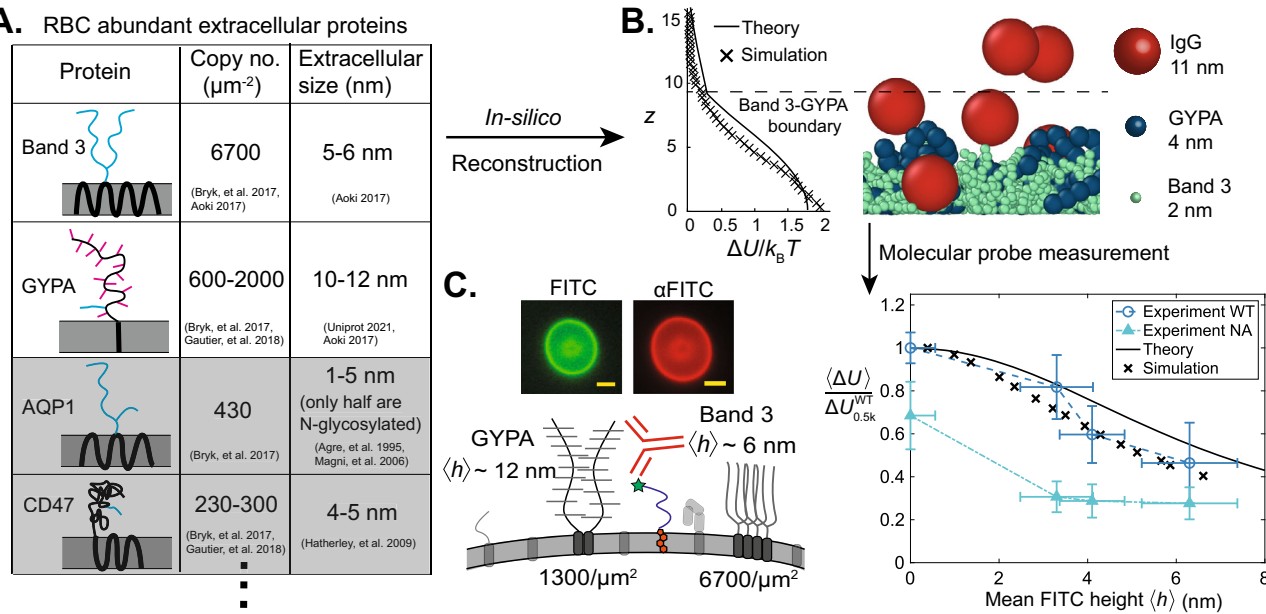

**Fig. 3 | Red blood cell (RBC) membrane proteomics is integrated into an in-silico model to predict surface crowding heterogeneity validated by theory and experiments. A** Relevant data from RBC proteomics for predicting surface crowding variation with extracellular height. Four of the most abundant RBC proteins with extracellular domains[35] are characterized according to copy number and extracellular domain size: Band 3, glycophorin A (GYPA), aquaporin 1 (AQP1), and CD47[35–37,101–104]. Simplified schematics of transmembrane and extracellular domains are sketched in a lipid bilayer, with black peptide sequences, blue N-glycans, and magenta O-glycans. GYPA and Band 3 are the only proteins of sufficient extracellular size and density to be considered in our model. **B** MD simulations and analytical theory describe steric repulsion between an IgG antibody and a model RBC glycocalyx. (Left) Repulsive energy penalty $\Delta U$ from both analytical theory and MD simulation is plotted as a function of height. (Right) Snapshot of MD simulations with spherical IgG adsorbing amongst course-grained Band 3 and GYPA.

**C** Molecular probes verify in-silico crowding heterogeneity predictions. (Upper left) Fluorescence micrographs of cholesterol-PEG-FITC and anti-FITC IgG binding to RBCs (scale bar 5 μm). (Lower left) Schematic of IgG binding to synthetic antigen sensors on a RBC surface, with GYPA and Band 3 dominating crowding. (Right) Mean crowding energy $\Delta U$, normalized by the surface value on wild-type (WT) human RBCs $\Delta U_{0.5k}^{WT}$ is plotted against mean cholesterol-PEG-FITC sensor height $\langle h \rangle$ for WT RBCs (blue circles) and RBCs treated with neuraminidase (NA, green triangles). Analytical theory (black curve) and MD simulation (black crosses) predict weaker free energy penalties of crowding with increasing antigen sensor height on WT cells. Dashed lines connecting experimental data are guides to the eye. Height data (x-axis) is CSOP data presented in Fig. 1. Experimental data points are derived from binding isotherm curve fits, and vertical error bars are derived as described in Fig. 2C. Curve fits are generated from populations of $n = 912, 905, 1062$, and 635 beads, for $K_{D,0}$ and populations of $n = 6113, 14476, 5680$, and 3335 RBCs for $K_D$.

density $P_{FITC}$ by the crowding penalty $\Delta U$ and integrating across all space:

$$\langle \Delta U \rangle = \int_0^\infty P_{FITC}(z; \langle h \rangle) \Delta U(z) dz. \qquad (2)$$

To describe $P_{FITC}$, we invoke the continuous Gaussian chain model of a surface-tethered polymer of mean height $\langle h \rangle$ in an ideal solvent, calculating the chain-end distribution (see Supporting Information for calculations)[34]. We verified $P_{FITC}$ with coarse-grained MD simulations of dilute surface-tethered PEG polymers, finding that the end-monomer distribution closely agrees with the theory (Fig. 2B, Supplementary Movie 1). Numerically evaluating the integral in Eq. (2) for a set of PEG-FITC sensors with mean heights $\langle h \rangle$ yields matching theoretical and computational predictions for the observed crowding profile as a function of mean sensor height (Fig. 2C).

Recasting the data from Fig. 1C in the form given by Eq. (1) and plotting as a function of the mean sensor heights reported in Fig. 1B, shows quantitative agreement with the theoretical and MD profiles developed in Eq. (2) (Fig. 2C). Our experimental and simulation data support a mechanism by which the brush sterically excludes the antibody, suggesting that our synthetic antigen sensors act as direct reporters of crowding heterogeneities with nanometer resolution.

### Synthetic sensors validate crowding predictions based on red blood cell proteomics

After validating our experimental antigen sensors on reconstituted membranes with analytical theory and coarse-grained simulations, we

sought to use theoretical and computational methods synergistically with experiments to map the extracellular crowding landscape of the human red blood cell (RBC). Since the RBC surface proteome is fully-characterized[35,36], we identified the most abundant extracellular proteins, and estimated extracellular domain sizes (Fig. 3A). In particular, we identified two abundant proteins with bulky extracellular domains: anion transporter Band 3 and mucin-like sialoglycoprotein GYPA[35–37].

Using both analytical theory and coarse-grained MD simulations, we modeled the RBC glycocalyx as a bidisperse polymer brush whose extracellular crowding profile opposes the adsorption of colloids like IgG (Fig. 3B, Supplementary Movie 3). We acknowledge that not all cell surface biopolymers can be represented as brushes; certain glycoproteins form gel-like meshes that can restrict colloidal transport[38,39]. However, while surface proteins may crosslink, to our knowledge the extracellular domains of GYPA and Band 3 have not been reported to do so. We used the lengths of extracellular peptides and glycans to estimate both the statistical monomer size and chain height of GYPA and Band 3 (see Supporting Information). We coarse-grained the biopolymers as simple bead-spring polymers that interact by excluded volume interactions and a bending penalty to account for additional effects like electrostatic repulsion that may alter the polymer persistence length. Our goal in the simulations is to develop a minimal model to capture the key qualitative trends of height-dependent crowding on a crowded cell surface, but additional effects like electrostatic interactions and crosslinking of surface biopolymers may be included to improve model accuracy. We input predicted chain height and known chain grafting densities into a model that superimposed two parabolic polymer brush density profiles[28], and applied the scaling $\Delta U \sim \phi^{9/4}$ to

model the repulsive energy penalty[22,32]. Note that this simplification treats the two brushes independently, with no interactions between the two species. Thus the curve (plotted in Fig. 3B) has a discontinuous slope at the point at which the monomer density in the shorter brush (Band 3) is predicted to reach zero. We also developed an in-silico model of a bidisperse brush, with each protein modeled as a bead-spring polymer (see Supporting Information for coarse-graining details). Fig. 3B shows close agreement between the analytical and MD descriptions of the glycocalyx, with the MD crowding energy likely decaying faster because it relaxes the assumption of a strongly-stretched brush inherent to the theory of Milner, Witten, and Cates[28].

To verify our predicted z-direction crowding profile, we incubated human RBCs in our synthetic antigen sensors so that the sensors incorporated to roughly equal surface concentrations (Supplementary Fig. S8). Unincorporated sensors were thoroughly washed from the bulk to prevent quenching of unbound antibody, with ~80% of sensors remaining bound over the course of the experiment (~1 hour). We measured the dissociation constant of anti-FITC binding to PEG0.5k, 2k, 5k, and 10k sensors, normalizing by the uncrowded $K_{D,0}$ on beads to find $\langle \Delta U \rangle$ (Eq. (1)). Antibody binding increased ≈5x from the most surface-proximal (PEG0.5k) to the most membrane-distal probe (PEG10k), corresponding to the crowding free energy penalty doubling from $z = 6.5$ nm to $z = 0$ (Fig. 3C). The experimental crowding landscape closely tracks the theoretical and simulated free energies, weighted by the FITC distributions in Fig. 2B.

As a control, we also treated RBCs with neuraminidase (NA), which cleaves negatively-charged sialic-acid from glycans exposed on the cell surface, confirming cleavage via a ≈60% reduction in wheat germ agglutinin binding on the cell surface (see supporting Fig. S9). We found a ≈30% reduction in crowding at the surface on NA-treated cells when compared to WT, as well as a flatter crowding profile for larger, more membrane-distal sensors (Fig. 3C). Given that GYPA contains ~75% of RBC sialic-acid, and that between one-third to one-half of sugars on its 15 O-glycosylations are sialic acid[36], this result suggests that GYPA plays a major role in mediating RBC crowding heterogeneity. This reduction in crowding is consistent with prior work by Takatori and Son et al. which showed similar reductions in crowding at the RBC surface ($h = 0$) using dextran-based sensors upon sialic acid cleavage, with NA treatment shortening the RBC glycocalyx mean height by about 30%[23]. These authors also showed simulations suggesting that the removal of charge may also play a role in de-swelling of the glycocalyx[23]. Given an approximate Debye length of 0.7 nm in phosphate-buffered saline, we expect that adjacent glycan-glycan charge interactions may play a role in de-swelling the polymer brush, but expect the charges to be largely screened at the length scales of glycan-IgG interactions.

These data demonstrate that for the relatively simple RBC plasma membrane, detailed proteomics data including copy number, structure, and glycosylation of surface proteins provide a robust approximation of membrane-orthogonal crowding heterogeneity. Computational techniques like machine learning are rapidly accelerating the identification of surface proteins and glycosylation sites[40,40,41], and with more detailed characterization of glycan sequences and surface protein densities on the horizon, we expect that the in-silico reconstruction of more complex mammalian cells will become feasible. Mapping crowding heterogeneities on these nanometer length scales with simulations and molecular probes may reveal the accessibility of receptors based on height, improving our understanding of signaling and optimizing drug delivery target selection.

### Development of phase-partitioning antigen sensors to measure lateral heterogeneities in surface crowding

Lateral heterogeneities in the composition of lipids, cholesterol, and proteins on plasma membranes, including lipid rafts and protein clusters, have been hypothesized to govern various physiological processes, like signal transduction, endocytosis, and membrane reorganization[42-51]. Leventhal et al. showed that ordered domains on giant plasma membrane vesicles (GPMVs) isolated from cells are depleted of transmembrane proteins[52-54] while Gurdap et al. further showed glycosylation and extracellular protein size to be inversely correlated with ordered domain partitioning in GPMVs[17], suggesting that crowding is likely reduced in more ordered domains like lipid rafts, compared to the bulk of the cell. However, while plasma membrane vesicles undergo mesoscopic phase separation, lipids and proteins on live cells are known to form transient 10–200 nm domains and clusters, which often exist on sub-second timescales[46,55-58]. As a result, the optical characterization of small, transient protein and lipid clusters on live cells is challenging[59-61]. To probe the lateral crowding heterogeneities that one might expect to arise from the lateral segregation of membrane constituents mammalian cells, we used different antigen sensors that either distribute approximately uniformly on the cell surface, or self-associate to form clusters with unique local protein and lipid compositions. By measuring IgG binding to these different antigens on both reconstituted and live plasma membranes, we then show that cell surface crowding can vary laterally on nanometer length scales.

In this section, we present crowding measurements on phase-separated reconstituted giant unilamellar vesicles (GUVs) where spatially heterogenous crowding is engineered to be easily visualized. We produced GUVs containing the ternary lipid mixture 2:2:1 1,2-dipalmitoyl-sn-glycero-3-phosphocholine (DPPC):DOPC:cholesterol, which phase separates into micron-scale liquid-ordered (Lo) and liquid-disordered (Ld) domains[62]. We preferentially crowded the Ld phase with 2% DOPE-PEG2k (Fig. 4A) and added DOPE-biotin and 1,2-dipalmitoyl-sn-glycero-3-phosphoethanolamine-N-(biotinyl) (DPPE-biotin) to present the biotin antigens in each phase. We measured the crowding energy for $\alpha$ Biotin IgG binding to each domain (Fig. 4A). Consistent with the experiments in Figs. 1–2, the PEG brush inhibited antibody binding on the crowded Ld domain and increased the normalized effective $K_D$ by 60% compared to the bare surface (Fig. 4B). In contrast, $\alpha$ biotin binding in the less-crowded Lo domain did not change relative to a bare membrane.

Although macroscopic phase domains on GUVs enable a simple measurement of lateral crowding heterogeneity, this approach is often impossible on live cell surfaces where heterogeneities can occur on diffraction-limited length scales. To address this challenge, we performed crowding measurements on SLB-coated beads with the same ternary lipid mixture, where the underlying substrate friction arrests phase domains into ≈90 nm nanoscopic features, similar to the size of lipid rafts and self-assembled protein clusters[46,58,63]. Since the individual phase domains cannot be identified, we measured the crowding on each phase by quantifying $\alpha$ biotin IgG binding on beads containing only one type of antigen: either DPPE-biotin or DOPE-biotin.

As shown in Fig. 4B, we found that the Ld antigen (DOPE-biotin) reported a crowding penalty 7× higher than the Lo antigen (DPPE-biotin). While the absolute magnitudes of observed $\Delta U$ were higher on beads than GUVs, we attribute this difference to lower incorporation of DOPE-PEG2k through the GUV electroformation process. Given the strong qualitative difference in crowding reported by antibody binding to antigens partitioning into Lo or Ld domains, we conclude that laterally-segregating antigen probes suitably report diffraction-limited lateral crowding heterogeneities.

### Antibody binding to surface-clustering antigens reports lateral crowding heterogeneities

Motivated by our ability to measure crowding heterogeneities on nanoscopic phase domains on reconstituted membranes, we used cluster-forming proteins as antigen probes to measure in-plane crowding heterogeneities on live mammalian cells. It has been shown in-vitro that when cholera toxin B (CTB) binds to the

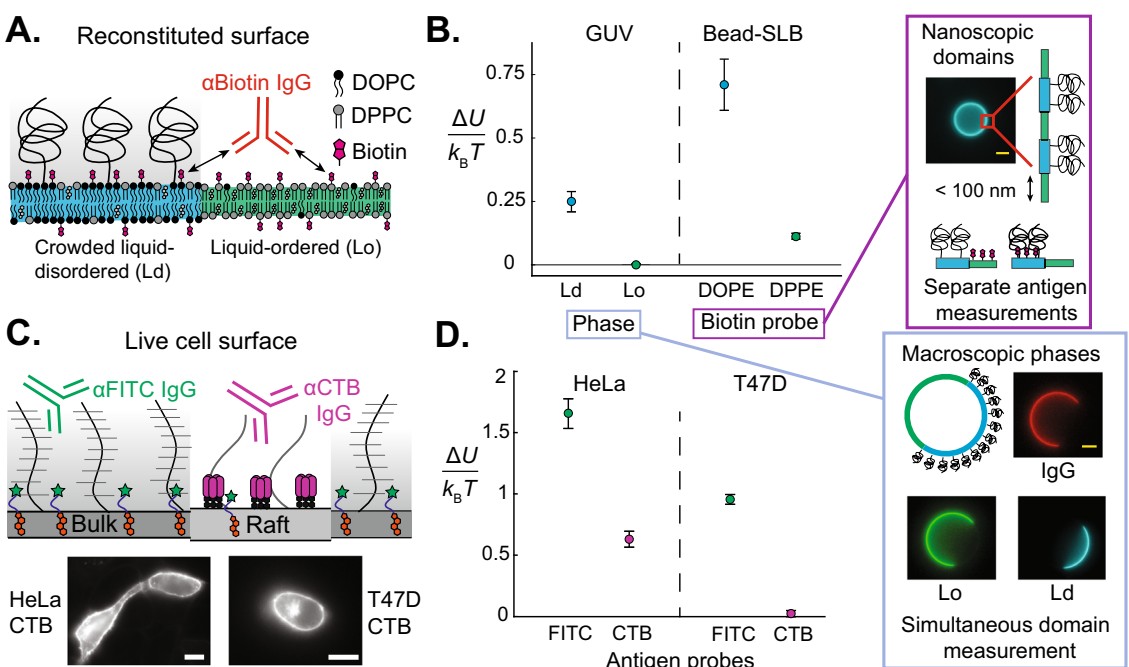

**Fig. 4 | Antibody binding reports lateral crowding heterogeneities on membranes with coexisting domains. A** Reconstituted ternary lipid mixtures of DOPC/DPPC/cholesterol containing 2% DOPE-PEG2k preferentially crowd the liquid-disordered (Ld) phase compared to the liquid-ordered (Lo) phase. DOPE- or DPPE-biotin antigens enable IgG antibody binding in the two phases. **B** Direct measurement of lateral crowding heterogeneity on macroscopic phase-separated domains in giant unilamellar vesicles (GUVs) yields similar trends to indirect crowding measurement of diffraction-limited domains on beads. (Left) The PEG brush crowding free energy is plotted for each domain on GUVs and for each antigen probe on kinetically-arrested bilayers on beads. (Lower right inset) We observed strong $\alpha$ Biotin IgG binding (red) only along the bare Lo surface (green), but not along the PEG-crowded Ld surface (blue) for the phase-separated GUVs (scale bar 5 μm). (Upper right inset) Ternary lipid bilayers on beads form nanoscopic domains due to the underlying substrate friction-arresting domain coarsening, so we measure crowding-free energy with only one antigen sensor type at a time. DOPE-biotin prefers Ld and DPPE-biotin prefers Lo, so their avidities report local crowding in their respective phases (scale bar 2 μm). **C** Antigen probes partition amongst different cell membrane domains, enabling direct characterization of lateral crowding heterogeneities on live cancer cells. (Upper) Cholera toxin B (CTB) antigens bind to ganglioside GM1, clustering and forming domains with a distinct protein/lipid composition, while cholesterol-PEG0.5k-FITC favors the disordered bulk membrane. $\alpha$ FITC or $\alpha$ CTB IgG binding reports lateral crowding heterogeneities between these domains. (Lower) Human cervical cancer HeLa and breast cancer T47D cell membranes labeled with CTB (scale bars 10 μm). **D** Crowding free energy reported by CTB and FITC sensors on HeLa and T47D cells. Error bars are derived as described in Fig. 2C. Populations of $n = 129$ GUVs, each, were used to find $K_{D,0}$ and 131 each for $K_D$. Populations of n=727 and 1005 beads were used to find $K_{D,0}$ and $n = 1177$ and 882 beads for $K_D$, for DOPE and DPPE respectively. Populations of $n = 126$ HeLa/FITC, 172 HeLa/CTB, 85 T47D/FITC, and 174 T47D/CTB cells were used.

ganglioside GM1, it can trigger the condensation of liquid-ordered lipid domains, enriched in CTB and GM1, from previously homogeneous lipid mixtures[64]. Not only are these 2D condensates enriched in CTB and GM1, but Hammond et al. showed that they can also exclude certain transmembrane proteins in vitro[64]. Other in vitro experiments have revealed that ganglioside-binding toxins can alter liquid-ordered domain composition and form self-associating clusters within phase domains[65]. We hypothesized that by leveraging the ability of CTB to form unique protein/lipid condensates on the surface, we could use anti-CTB antibody binding to probe crowding in a unique extracellular microenvironment, and thus gain insight into lateral crowding heterogeneities on live cells.

We bound Alexa Fluor 488-labeled CTB to HeLa human cervical cancer cells and T47D human breast cancer cells and measured crowding by comparing $K_D$ of $\alpha$ CTB IgG to the intrinsic binding affinity of $\alpha$ CTB on beads. CTB bound strongly to both cell lines, with approximately 40% greater binding on HeLa than on T47D (Fig. 4C, Supplementary Fig. S10). We compared the $K_D$ of $\alpha$ CTB IgG to that of $\alpha$ FITC binding to cholesterol-PEG0.5k-FITC sensors to differentiate CTB cluster-specific crowding from that of the bulk cell membrane (Fig. 4C). GM1 protrudes only 1–2 nm above the bilayer surface[66] and CTB is only ~2–3 nm in size[67], so we assume that crowding at the CTB epitope is similar to that at the bilayer surface. While cholesterol-PEG-FITC has been observed to show a slight preference for ordered domains in GPMVs (26% greater enrichment in ordered domains)[68], we

observed nearly uniform partitioning of cholesterol-PEG0.5k-FITC sensors between Lo and Ld phases on GUVs (Supplementary Note 3, Supplementary Fig. S11), which is consistent with other work showing that functionalized cholesterol tends to favor the disordered phase due to its reduced packing efficiency[49,69]. In addition, Windschiegl et al. showed that ganglioside-binding toxins can exclude Lo-favoring fluorescent dyes into the Ld phase on reconstituted bilayers[65]. Given these findings, we expect that any cholesterol-PEG-FITC enrichment in CTB/GM1 clusters is modest compared to that of CTB, and thus consider crowding reported by cholesterol-PEG0.5k-FITC to be approximately representative of the mean cell surface crowding.

We measured spatial crowding heterogeneities on human cervical cancer HeLa and human breast cancer T47D cells, both of which have surface proteomes rich in bulky proteins[6,70]. Shurer et al. showed that wild type cells of both types form membrane tubules in response to high surface crowding[6]. The crowded surfaceomes of HeLa and T47D cells make these cells rich models for studying lateral heterogeneities. Bulk crowding for both cell lines as measured by our cholesterol-PEG0.5k-FITC sensors was on the order of 1–1.5 $k_B T$, consistent with the brush exclusion energy on the surface of RBCs (Fig. 4D).

On both cell lines, CTB antigens reported significantly less crowding than the cholesterol-PEG-FITC antigens, suggesting that the extracellular space around CTB is not heavily crowded with proteins and sugars (Fig. 4D). Moreover, less CTB bound to the surface of T47D than HeLa, but more $\alpha$ CTB IgG bound to T47D (Supplementary

Fig. S10), offering further evidence that any reduction in IgG binding is due to local environmental factors, such as crowding. Between the two cell lines, the ratio of bulk to raft crowding free energy is also 18x greater in T47D than in HeLa, suggesting that the composition of GM1-enriched domains is cell-specific, and that the lateral distribution of bulky extracellular proteins may vary considerably amongst different cell lines. Our unique ability to probe native cell membranes will advance further mechanistic insight into the roles of the actin cytoskeleton and other structural complexities on glycocalyx organization.

The reduction in crowding induced by CTB/GM1 clusters is qualitatively consistent with the results of Levental et al.[52–54] and Gurdap et al.[17] and supports the hypothesis that more ordered lipid domains of native membranes exclude proteins that contribute to extracellular crowding. This is a significant result because GPMVs exclude membrane proteins that are bound to the actin cytoskeleton, and it has long been unclear whether actin, myosin, and other structural features affect surface crowding. Our results reflect the first measurement to our knowledge of nanometer-scale crowding heterogeneity on an live plasma membrane, with intact cytoskeletal dynamics.

However, due to the fact that CTB both forms clusters and influences the local lipid composition, we cannot directly extrapolate our reduced CTB crowding to be representative of lipid rafts in general. CTB is known to drive GM1 association with caveolar invaginations in the membrane, and its lateral diffusion is restricted in an ATP-dependent manner that is not the norm for the GPI-anchored proteins that normally associate with rafts[44,71]. Therefore, we conducted an additional control experiment to determine whether protein clustering alone is sufficient to alter the crowding landscape of the cell. We incubated HeLa in FITC-conjugated annexin V, a protein that binds to phosphatidylserine (PS) and forms a 2D lattice of repeating hexameric clusters on the membrane[58,72]. PS is typically restricted to the intracellular leaflet of the plasma membrane on healthy cells, and thus our measurements were largely confined to blebbed regions of the membrane, in which PS was expressed the extracellular leaflet (see Supplementary Fig. S12)[60]. Here, we found similarly reduced crowding to CTB (Supplementary Fig. S12), suggesting that simply clustering antigen proteins together may be sufficient to exclude other bulky extracellular proteins locally, leading to lateral variations in crowding.

## Discussion

In this work, we developed a simple experimental technique to study the spatial heterogeneities of surface crowding on live cell membranes with exquisite spatial resolution. Alternative approaches like detergent-resistant membranes (DRMs) and GPMVs[50] are invasive techniques that do not provide a description of surface organization in the native cell membrane environment. Prior to this work, existing techniques were capable of measuring spatial organization on cell surfaces with a very thick glycocalyx (0.2-1 $\mu$ m), such as endothelial cells in the vasculature[73–76]. However, studying the spatial organization of live cell surfaces with glycocalyx thicknesses of ~ 10 nm was a challenge because standard optical microscopy cannot resolve nanometer variations.

While not reporting spatial heterogeneity, recent measurements with membrane-binding macromolecular probes reported osmotic pressures of 1–4 kPa at the surface of mammalian cells[23]. These surface pressures are comparable to and in some cases larger than the stiffness of the cell cortex (≈1 kPa), providing new insight on the physical role of protein and glycan crowding on cell membranes. In this work, we demonstrated that these pressures are highly dependent on proximity to the membrane surface and that glycocalyx crowding decays rapidly away from the membrane. Our probes are physiologically relevant because many protein-protein interactions occur at a finite distance away from the membrane, like kinetic segregation in T-cell receptor triggering (≈ 10–15 nm)[77,78].

We present our antigen sensors on live cell surfaces with nanometer precision and use antibody binding equilibria to directly report spatial variations in surface crowding. Our sensors achieve this nanometer spatial sensitivity by leveraging the exponential amplification of our readout ($K_D$) as a function of the crowding energy, $K_D \sim \exp(\Delta U/(k_B T))$ (Eq. (1)). This exponential amplification distinguishes our approach from previous techniques that rely on polymer brush height as the readout of crowding, which scales weakly with surface density, $h \sim n^{1/3}$[19,22,32]. Taking the energy barrier to be proportional to the osmotic pressure, and in turn surface density, $\Delta U \sim \Pi \sim n^{9/4}$[22,32], we obtain the scaling $h \sim (\Delta U/(k_B T))^{4/27}$, which is significantly weaker than $K_D \sim \exp(\Delta U/(k_B T))$. The ≈1$k_B T$ change in crowding we observe within 6 nm of the RBC surface confirms this spatial sensitivity, and demonstrates the power of our technique in characterizing the highly heterogeneous membrane-proximal surfaceome, in which surface signaling and viral entry occur[30,31].

Monoclonal antibody (mAb) drug candidates are currently screened using surface-plasmon resonance (SPR), in which binding affinity and avidity are measured on a bare hydrogel chip without regard to multi-body interactions[79]. With mAbs like the breast cancer treatment trastuzumab targeting a 4 nm tall epitope on human epidermal growth factor receptor 2 (HER2), probing local crowding variations may inform target selection and improve potency[31]. Indeed, Chung et al. found that trastuzumab and pertuzumab attenuate tubule structures enriched in HER2, suggesting that biophysical interactions like crowding may influence the potency of mAb therapies[80]. Our crowding measurements may also help inform the biophysical mechanisms governing antibody-dependent phagocytosis, which have been recently shown to have a strong dependence on the relative heights of the macrophage Fc$\gamma$ receptor and the target cell surface proteins[13,81]. In conclusion, our sensors may be used to inform important physiological processes, like antibody binding to buried surface receptors, membrane organization of lipid raft-like domains, and cellular phagocytosis.

When characterizing the RBC surface, we also demonstrated the potential to augment experimental crowding measurements with an in-silico cell surface reconstruction based upon proteomics data. As recent advances in surface proteomics continue to better characterize glycocalyx components for a broad host of cell lines[40,40,41], we expect that accurate in-silico model will become possible on more complex mammalian cell surfaceomes. A technology to describe the extracellular crowding landscape for any cell a priori, using only proteomics data, may advance our basic understanding of cell membrane biology.

Using laterally-segregating antigen probes that exist in distinct crowding microenvironments on the plasma membrane, we demonstrated reduced crowding in GM1/CTB-enriched clusters on T47D and HeLa cells. These findings are consistent with the known reduction in transmembrane protein density on ordered domains in GPMVs[52–54], but our noninvasive measurements on live cells provide further insight into the dynamic cell surface ecosystem, including the interplay between the actin cytoskeleton and the membrane. Indeed, there has been considerable interest in the connection between the cytoskeleton and transmembrane protein organization over the past few decades[82–84], as actin is known to redistribute lipids and proteins on the cell surface[60,61,85,86]. By bridging this gap and characterizing CTB-enriched clusters on live cells, we speculate that on length scales of order 10–100 nm, the cytoskeleton may not dramatically change bulky protein composition beyond that of equilibrium domains, supporting the use of actin-free systems like GPMVs to measure the lateral distribution of crowders[17,52].

Viral particles like simian virus (SV) 40 and other polyomaviruses[87,88], and toxins like Shiga and cholera toxin[89,90], bind to gangliosides like GM1. SV40 virus is ≈ 45nm in size[91], about three times larger than an IgG. Since the mechanical work required to insert a particle into a crowded space scales approximately as the

particle volume (see Supporting Information), a viral particle would be posed with an energy barrier of $\Delta U_{virus} \sim \Delta U_{IgG}(R_{virus}/R_{IgG})^3 \approx 20-30 k_B T$ if it tried to penetrate the glycocalyx above the bulk membrane of T47D cells. In contrast, in ganglioside-enriched domains like those we study in this work, the binding penalty is merely $\Delta U_{virus} \approx 0.5-1 k_B T$ on T47D cells, suggesting that viral particles may experience a thirty-fold larger effective affinity towards the less-crowded, ganglioside-rich domains.

However, the discrepancy between relative raft-to-bulk crowding on HeLa and T47D cells indicates that the effect of GM1/CTB-enriched domains in reorganizing the glycocalyx varies considerably from cell to cell. Future direct comparisons between lateral heterogeneity on both live cells and their secreted membrane vesicles will provide a more thorough description of the fraction of extracellular bulk that remains anchored to the cytoskeleton in both disordered and raft-like membrane domains.

## Methods

### Antigen probe synthesis
Cholesterol-PEGx-NH2, where x represents PEG0.5k, 2k, 5k, or 10k, was reacted with a 10× excess of N-hydroxy-succinimidyl ester (NHS)-FITC, overnight at 50 °C in dimethylsulfoxide (DMSO). Unreacted FITC was removed via a 7K MWCO Zeba spin desalting column. SLB-coated beads were incubated with 100 nm FITC antigen sensors for 15 minutes at room temperature.

### Microscope for all imaging experiments
All imaging was carried out on an inverted Nikon Ti2-Eclipse microscope (Nikon Instruments) using an oil-immersion objective (Apo ×60, numerical aperture (NA) 1.4, oil; Apo ×100, NA 1.45, oil). Lumencor SpectraX Multi-Line LED Light Source was used for excitation (Lumencor, Inc). Fluorescent light was spectrally filtered with emission filters (432/36, 515/30, 595/31, and 680/42; Semrock, IDEX Health and Science) and imaged on a Photometrics Prime 95 CMOS Camera (Teledyne Photometrics). Microscope images were collected using MicroManager 1.4 software[92].

### Sensor height measurement
Small unilamellar vesicles (SUVs) were formed using an established sonication method[13]. A lipid film containing 1,2-dioleoyl-sn-glycero-3-phos-phocholine (DOPC), 3% 1,2-dioleoyl-sn-glycero-3-phosphoetha-nolamine-N-[methoxy(polyethylene glycol)-2000] (DOPE-PEG2k), and DOPE-rhodamine was dried under nitrogen and then vacuum for 30 minutes. The film was rehydrated in Milli-Q (MQ) water to 0.2 mg/mL lipids, sonicated at low power using a tip sonicator (Branson SFX250 Sonifier) at 20% of maximum, 1s/2s on/off, for three minutes. We added MOPS buffer at a final concentration of 50 mM MOPS pH 7.4, 100 mM NaCl to the resulting SUV mixture. Then, 10 μL of 4 μm silica bead slurry (10% solids) was cleaned with piranha solution (3:2 $H_2SO_4$:$H_2O_2$) and washed three times with 1 mL MQ water before being suspended in 100 μL MQ water (1% solids). 3 μL of bead slurry was mixed with 30 μL SUVs and incubated for ten minutes at room temperature before washing five times with HEPES buffer (50 mM HEPES pH 7.4, 100 mM NaCl).

FITC sensor heights were established using CSOP[31]. SLB-coated beads were incubated in 200 nm cholesterol-PEGx-FITC at room temperature for 15 minutes, where x represents PEG0.5k, 2k, 5k, and 10k. Unbound sensors were washed from the bulk and CSOP measurement used to find the difference in apparent bead radius on the 488 nm FITC channel and 555 nm rhodamine channel $\langle h_{observed} \rangle$. To correct for chromatic aberration, a baseline difference in 488 nm and 555 nm radii $\langle h_{baseline} \rangle$ was measured on SLB-coated beads containing DOPC with 0.05% DOPE-rhodamine, and 0.05% DOPE-Atto 488 lipids. The FITC antigen height was obtained by subtracting this baseline from the observed height: $\langle h \rangle = \langle h_{observed} \rangle - \langle h_{baseline} \rangle$.

### Dissociation constant measurement for reconstituted PEG brushes
4 μm SLB-coated beads with PEG brushes were formed using a mixture of DOPC, 3% DSPE-PEG2k, and 0.05% DOPE-rhodamine. Bare beads for measuring $K_{D,0}$ were formed with only DOPC and 0.05% DOPE-rhodamine. Beads were incubated in 100 nm cholesterol-PEGx-FITC antigen sensors for 15 minutes at room temperature, then washed with HEPES buffer.

Lysine residues of anti-FITC (α FITC) IgG antibodies were randomly labeled by reacting with 8x excess NHS-Alexa Fluor 647 for one hour at room temperature in 50 mM sodium bicarbonate solution. Unreacted dye was separated via a 7MWCO spin desalting column and the recovery and labeling ratio measured via Nanodrop UV-vis spectroscopy.

Coverslips were passivated with 1 mM bovine serum albumin (BSA) to prevent nonspecific antibody adsorption. Antigen-coated beads were added to coverslip wells containing α FITC-647 and allowed to sediment and equilibrate with IgG for 30 minutes at room temperature. Bulk antibody concentrations ranged from 0.67 to 20 nm (see Source Data). At least 50 beads were imaged for each bulk IgG concentration, with an approximately equatorial focal plane. Images were subdivided into individual beads, and the edges identified by the brightest 5% of pixels, on the 555 nm (DOPE-rhodamine) channel, for each subimage. The background intensity was taken to be the 30th percentile of α FITC intensities for each bead subimage, and the bead intensity signal was calculated by subtracting background from the α FITC signal associated with the brightest rhodamine pixels. The intensity signal for each bead was averaged to yield a mean bead signal, and the mean bead signals were then averaged for each α FITC bulk concentration, and fit to a Hill isotherm to find $K_D$.

### Dissociation constant measurement for CD45 antigens
A GYPA brush with dilute CD45 antigens was reconstituted by incubating beads in 10:1 GYPA:CD45. SLB-coated beads containing DOPC, 8% 1,2-dioleoyl-sn-glycero-3-[(N-(5-amino-1-carboxypentyl) iminodiacetic acid)succinyl] (DGS-Ni-NTA), and 0.2% DOPE-Atto 390 were incubated with 10 nm His-tagged mouse CD45 and 100 nm His-tagged glycophorin A (GYPA) for 15 minutes at 37 °C. Unbound protein was washed five times from the bulk with HEPES buffer. GYPA was labeled with NHS-Alexa Fluor 555 and CD45 was labeled with NHS-Alexa Fluor 488, and we thus confirmed a qualitative excess of the GYPA blockers on the beads. Beads were incubated in either Alexa Fluor 647-labeled α C363 or α I3 on CD45 for 30 minutes, and the $K_D$ measured. Antibody bulk concentrations ranged from 0.33 to 20 nm for α C363 and 0.67 to 33 nm for α I3 (see Source Data). Baseline $K_{D,0}$ for was measured for both CD45 epitopes on beads with no GYPA.

GYPA height was measured via CSOP using fluorescently-labeled purified anti-human glycophorin AB monoclonal antibody (clone HIR2), BioLegend (cat. no. 306602). Antibody was diluted to 10 nM.

### Red blood cell (RBC) dissociation constant measurement
Single-donor human whole blood in K2-EDTA was purchased from Innovative Research and used within three days of arrival. The researchers in this study had no contact with human subjects and vendor samples were de-identified, precluding a need for IRB clearance. Blood was centrifuged at 300g for five minutes to isolate RBCs. For experiments with sialic-acid deficient red blood cells, red blood cells were treated with 100 mU/mL neuraminidase at 37 °C for two hours. Cells were centrifuged and washed with PBS three times to remove bulk neuraminidase. RBCs were incubated with 100 nm cholesterol-PEGx-FITC (x represents PEG0.5k, 2k, 5k, and 10k) antigen sensors for 15 minutes at 37 °C. RBCs were diluted ×10 in PBS, centrifuged, and the supernatant discarded to remove unbound cholesterol. RBCs were washed in this way four more times, with a ~100× dilution in fresh PBS each time. RBCs were added to α FITC-647,

pipette mixed, and incubated for 30 minutes before imaging. Antibody bulk concentrations ranged from 0.13 to 27 nm (see Source Data). RBC images were analyzed using the same methods as beads, with at least 50 cells per IgG concentration, to calculate $K_D$. For $\Delta U$ calculations, $K_D$ was normalized against the bare-bead $K_{D,0}$, for each antigen.

## Lateral crowding heterogeneity measurements on mesoscale membrane domains

Antibody dissociation constants were measured on crowded and uncrowded coexisting domains in liquid-liquid phase-separated giant unilamellar vesicles (GUVs). DOPC, 1,2-dipalmitoyl-sn-glycero-3-phosphocholine (DPPC), and cholesterol were combined in a 2:2:1 ratio, which phase separates at room temperature[62]. 0.3% DOPE-biotin and 0.05% 1,2-dipalmitoyl-sn-glycero-3-phosphoethanolamine-N-(biotinyl) (DPPE-biotin) were added as liquid-disordered and liquid-ordered antigen probes, respectively. We set the relative amounts of DOPE- and DPPE-biotin such that the antigen density in each phase was approximately equivalent, as reported by $\alpha$ Biotin binding. GUVs were formed either with or without 2% DOPE-PEG2k, which formed a crowding brush in only the liquid-disordered (Ld) phase. 0.05% each of DOPE-rhodamine and 1,2-distearoyl-sn-glycero-3-phosphoethanolamine-N-[poly(ethylene glycol)2000-N′-carboxyfluorescein] (DSPE-PEG2k-FITC) were also added to label the Ld and liquid-ordered (Lo) phases, respectively.

GUVs were produced via a modified electroformation protocol[93,94]. Lipids dissolved in chloroform were spread onto an indium tin oxide (ITO)-coated slide and the resulting film dried under vacuum for greater than 30 minutes. The lipid film was rehydrated in a 300 mM sucrose solution, placed in contact with a second ITO-coated slide, and an AC sinusoidal voltage applied across the two slides: 10 Hz/ 1.0 V for two hours then 0.4V/2 Hz for 20 minutes. GUVs were electroformed at 50 °C to ensure phase mixing, then cooled below the melting point to room temperature once electroformation was stopped.

Phase-separated GUVs were incubated in Alexa Fluor 647-labeled $\alpha$ Biotin antibody for at least 30 minutes. Antibody bulk concentrations ranged from 2 to 67 nm (see Source Data). The GUV size distribution spanned tens of microns, requiring that each vesicle be imaged individually to preserve a consistent equatorial focus. For each vesicle, Lo and Ld domains were identified by selecting the brightest 2% of pixels on the 488 (DSPE-PEG2k-FITC) and 555 nm (DOPE-rhodamine) channels, respectively, for each GUV image. The corresponding 647 nm intensities for these pixels were averaged and subtracted from the bottom 30th percentile of 647 intensities across the entire image, yielding a mean intensity for each phase. Lo and Ld intensities were averaged across all GUVs for each bulk IgG concentration and fit to a Hill isotherm to find $K_D$ (with DOPE-PEG2k) and $K_{D,0}$ (without DOPE-PEG2k).

## Lateral crowding heterogeneity measurements on diffraction-limited domains

Antibody dissociation constants were measured for Lo- and Ld-favoring antigens on phase-separated SLBs, with kinetically-arrested nanoscopic domains. Beads were coated with SLBs containing 2:2:1 DOPC:DPPC:cholesterol, with 0.05% DOPE-rhodamine Ld label and 1% DOPE-PEG2k Ld crowder. Only one of either DOPE-biotin or DPPE-biotin was also included, localizing the antigens primarily on either the Ld or Lo domains, respectively. Beads were incubated in $\alpha$ Biotin-647, imaged, and $K_D$ fit for each antigen. $K_{D,0}$ was measured with SLBs containing no DOPE-PEG2k.

## Lateral crowding heterogeneity measurements on human cancer cells

Human cervical cancer HeLa and breast cancer T47D cells were obtained from ATCC. HeLa cells were cultured in Dulbecco's Modified

Eagle Medium (DMEM) and T47D cells were cultured in RPMI 1640 Medium, both supplemented with 10% fetal bovine serum (FBS) and 1% Pen-Strep. Cells were incubated at 37 °C with 5% $CO_2$.

Cells were plated approximately 24 hours before imaging. Immediately before imaging, media was exchanged with PBS containing either 100 nm cholesterol-PEG0.5k-FITC and cholesterol-PEG0.5k-Alexa Fluor 555; 20 nm annexin V-FITC; or 100 nm Alexa Fluor 488-labeled cholera toxin B (CTB), and incubated for 20 minutes at 37 °C. Unbound antigen was washed from the well five times with PBS.

Cells were incubated in either $\alpha$ FITC-647 or $\alpha$ CTB-647 for 30 minutes. Antibody concentrations ranged from 0.33 to 100 M (see Source Data). At least 15 cells were analyzed for each bulk concentration. Fluorescence images of individual cells were captured, focusing on the equatorial plane, so that the plasma membrane outline was clearly visible. To select pixels for analysis, we took the product of the antigen and antibody signal for each pixel, identifying the top 7% for analysis. We took the mean IgG signal for these pixels, for each cell, to obtain the peak signal for that cell. In HeLa and T47D cell measurements, we observed some sensor internalization into the cell interior, but the antibody largely remained on the exterior of the cell. Taking the product of IgG and antigen ensured that only plasma membrane signal was analyzed. For cells with cholesterol-PEG0.5k-FITC antigens, we used the co-incubated Alexa Fluor 555 constructs to identify the pixels of highest antigen density, because the $\alpha$ FITC IgG quenches FITC. The background signal was set to the bottom 30th percentile of pixels, and the mean of the differences between peak and baseline for each cell taken to represent the surface-bound antibody. Bound antibody fraction was plotted against bulk IgG concentration and fit using the Hill isotherm ($n = 2$) to find $K_D$.

The bare dissociation constant $K_{D,0}$ for $\alpha$ CTB was measured on SLB-coated beads containing DOPC and 0.05% ovine brain GM1. Beads were incubated with 100 nm CTB for 15 min, washed, incubated in $\alpha$ CTB for one hour, and then imaged. $K_{D,0}$ was fit to IgG intensity data according to the procedures in earlier bead experiments. For the cholesterol-PEG0.5k-FITC antigen, the $K_{D,0}$ value from earlier bead experiments was used. We calculated the free energies associated with CTB-reported raft-like domain crowding and FITC-reported bulk crowding using Eq. (1).

## Molecular dynamics simulations

To validate theoretical predictions for the surface crowding profile, we performed coarse-grained molecular dynamics simulations using a graphics processing unit (GPU)-enabled HOOMD-Blue simulation package[95,96]. We simulated membrane-bound PEG-conjugated FITC sensors using the Kremer-Grest bead-spring model[97] for polymers chains, with bead diameter $\sigma = 0.33$ nm to represent the ethylene glycol monomer. One polymer end was confined to the bottom of the simulation box using wall potentials but was allowed to diffuse laterally[31]. We imposed periodic boundary conditions along $x$ and $y$ while the $z = \pm L_z/2$ boundaries were impenetrable. We used a system box size of $V = L^2 L_z$ where $L_z = 50 - 200\sigma$ and $L$ was adjusted to achieve the specified surface density and number of chains. All particle pair interactions and wall potentials are modeled using the Weeks-Chandler-Anderson potential[98]. The bond potentials were modeled using the finite extensive nonlinear elastic (FENE) potential with spring constant $k = k_B T/\sigma^2$. The semiflexibility of polymer chains was imposed through a harmonic angle potential $U_B = \epsilon_B(1 - cos(\theta_{ijk} - \theta_0))$, where $\theta_{ijk}$ is the bond angle between adjacent particles ($i, j, k$), $\theta_0$ is the resting angle, and $\epsilon_B = k_B T L_P/L_B$ is the bending energy, defined with persistence length $L_P = 0.97\sigma$ and bond length $L_B = \sigma$. We first simulated the experimental surface density of ~1000 chains/$\mu m^2$ and averaged over ~2000 polymers to verify that chains were dilute and non-interacting. We then simulated single chains and varied the degrees of polymerization to span PEG0.5k to PEG10k. Using simulation snapshots, we

binned the spatial distribution of the FITC sensor normal to the surface, $P_{FITC}$. Single-chain dynamics were averaged over 15 simulations of 1000 snapshots each.

To characterize surface crowding, we separately simulated spherical antibody particles in the presence of surface-confined polymers. PEG2k crowders on reconstituted beads were modeled as a monodisperse polymer brush with degree of polymerization $N = 45$ and surface density of 30,000/µm$^2$ and averaged over ~1000 chains. In separate simulations, we also modeled RBC cell surface proteins using a bidisperse polymer brush with the same coarse-graining as PEG. GYPA was coarse-grained into a 7-bead chain with a bead diameter of 4 nm, corresponding to the size of the sugar side chains along the backbone. Band 3 was coarse-grained into a 10-bead chain with 2 nm beads, representing the two large branches of the N-glycan. We chose surface coverages of 1300 and 6700 chains/µm$^2$ to match reported copy numbers of GYPA and Band 3. The Fab region of IgG was coarse-grained into a single spherical bead of size 4 nm in simulations of the reconstituted PEG2k brush, while the full IgG antibody was coarse-grained into an 11 nm bead in simulations of the RBC surface. The 2–3 nm PEG2k brush is smaller than a ~10 nm IgG, so we assume only the Fab domain penetrates the reconstituted brush, while the full IgG penetrates the thicker RBC glycocalyx[99,100].

We calculated the probability distribution of antibodies on the cell surface in the presence of crowding polymers or proteins, and used the Boltzmann relation to compute the repulsive energy penalty $U_{brush}(z) = -\ln(P_{IgG}(z)/P_{bulk})$ of the brush at equilibrium. We numerically integrated Eq. (2) to compute the mean crowding energy $\langle \Delta U \rangle$ given height fluctuations in the FITC sensor (Figs. 2C, 3B).

### Reporting summary

Further information on research design is available in the Nature Portfolio Reporting Summary linked to this article.

## Data availability

All data generated in this study are provided in the main text, Supplementary Information, or Source Data file. Source data are provided with this paper.

## Code availability

Cell surface optical profilometry measurements were made using code from ref. 31, which is available at https://github.com/smson-ucb/CSOP. Simulations were conducted using HOOMD blue, which is available at https://github.com/glotzerlab/hoomd-blue.

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

## Acknowledgements

This material is based upon work supported by the National Science Foundation under Grant No. 2150686. D.P.A. is supported by the National Science Foundation Graduate Research Fellowship under Grant No. 2139319. Y.X. is supported by the Dow Chemical Discovery Fellowship in the Department of Chemical Engineering at the University of California, Santa Barbara. S.C.T. is supported by the Packard Fellowship in Science and Engineering. The authors acknowledge the assistance of Dr. Jennifer Smith, manager of the Biological Nanostructures Laboratory within the California Nano-Systems Institute, supported by the University of California, Santa Barbara. Use was made of computational facilities purchased with funds from the National Science Foundation (CNS-1725797) and administered by the Center for Scientific Computing (CSC). The CSC is supported by the California NanoSystems Institute and the Materials Research Science and Engineering Center (MRSEC; NSF DMR 1720256) at UC Santa Barbara.

## Author contributions

D.P.A. and S.C.T. conceived of the study; all authors designed research; D.P.A. performed experiments; Y.X. performed simulations; S.C.T. supervised the study; and all authors wrote the paper.

## Competing interests

The authors declare no competing interests.
