## [Peer Review File · Nature Communications]

Antibody binding reports spatial heterogeneities in cell membrane organizationREVIEWER COMMENTS

Reviewer #1 (Remarks to the Author):

In this manuscript, Arnold, Xu, and Takatori apply their unique methodology for nanoscale measurements of distance from a membrane surface to build a series of probes for measuring the impact of crowding on antibody-antigen binding. The probes are first tested in synthetic systems, then a theoretical model and coarse-grained simulations are used to describe and validate the approach. The probes are then applied to the surface of red blood cells and tumor cell lines to test the impact of endogenous crowding molecules on antigen detection. One key conclusion is a difference in crowding between different membrane domains on the cell surface.

Overall, the manuscript is well written, well-reasoned, and adds important evidence to an accumulating body of research on the biophysical impacts of the cell's boundary layer on the activity of molecules at the cell surface. While I am supportive of publication, I believe the manuscript could be improved by the following considerations:

1. While Fig1 uses a series of probes where the longest probe exceeds the height of the crowding molecule (10k = 7 nm, PEG2k crowder = ~4 nm), the same is not true for RBCs in Fig 3. A longer probe should be tested such that the probe height exceeds the expected GYPA height so that parity between crowded and non-crowded antibody binding can be demonstrated. Alternatively, cleavage or removal of GYPA could be used to show how modifying the crowding of RBCs impacts antibody binding
2. Cholera toxin is used as a raft-domain marker on cell lines to demonstrate that raft domains are less crowded than bulk membrane. However, glycolipid binding toxins can self-associate to form oligomers on the surface, including invaginations (see work by Johannes). Such cell surface domains may be quite tightly packed with toxin and exclude other proteins independent of the underlying lipid composition. A more direct way to assay raft dependent behavior may be to measure binding to a monomeric raft marker such as a GPI-anchored protein. Put another way, data in Fig 4C, D supporting the claim that "raft-like membrane domains exclude bulky membrane proteins and glycoproteins" rely on the assumption that CTB is marking raft domains and is not producing its own organization. Muc1 (which has been reported to be present in detergent resistant membranes) could still be present in raft domains, just not the same raft domains that underly CTB clusters.
3. To make a stronger argument that Muc1 is the main crowding actor on tumor cell lines, analogous experiments to those in Fig 4D but without Muc1 should be performed. This could be accomplished by cell sorting using anti-Muc1 antibodies or PNA to separate high and low/no Muc1 cell populations as in Paszek et al. Another suggestion is to cleave cell surface mucin with a commercially available enzyme (Mucinase StcE, MilliporeSigma #SAE0202).

Minor recommendations/comments:

- In my opinion, the current title is not sufficiently informative. For the sake of future readers, I suggest a more specific, descriptive title
- The authors report a predicted GYPA height for the experiments in Fig 1C. Since GYPA is recombinant in these experiments, they could add a tag at the “top” and directly measure the height of GYPA.
- More discussion of the small discrepancy between theory and simulations in Fig 3B would be helpful.
- The authors assume that their chol-PEG-FITC probes are equally distributed in raft and non-raft domains at the plasma membrane based on partitioning experiments in GUVs. This assumption may be questioned because the lipid composition of GUVs does not fully represent PM composition, which may impact chol-PEG-FITC partitioning. GPMVs are a better model to report chol-PEG-FITC partitioning for raft-like PM domains and have been reported to show slight enrichment of chol-PEG 2k-FITC in raft domains (<https://doi.org/10.1101/2022.08.02.502487>).
- Consider referencing Hammond et al PNAS 2005 which shows the coalescence of large membrane domains upon GM1 crosslinking by cholera toxin.
- Typo in reference 33 – Dan Fletcher listed three times.
- Line 79 – The Levental lab has reported the exclusion of bulky transmembrane domains from raft-like domains on GPMVs, but not whether there is an effect of cytoplasmic or extracellular domain size. However, Gurdap et al Biophys J 2022 has shown a decrease in ordered domain partitioning with increasing ectodomain mass/glycosylation, which would make a good reference here.

Reviewer #2 (Remarks to the Author):

Fig. 1. How much cholesterol-PEGx-FITC was incorporated to the beads? Is the beads washed after incubation with cholesterol-PEGx-FITC? Were all cholesterol-PEGx-FITC incorporated equally? How about the amount of free cholesterol-PEGx-FITC? How much cholesterol-PEGx-FITC was detached during incubation? How about the concentration dependence of DOPE-PEG2k on the antibody binding? Does GM1/CTB show similar results with cholesterol-PEGx-FITC/IgG?

Fig. 3. Similarity between simulation and experiment does not exclude the possibility that GYPA and Band 3 are major contributors of the inhibition of antibody binding. Cells without GYPA and/or Band 3 or deficient glycosylation need to be checked.

Fig. 4. Fig. 4C Fluorescence image of T47D cells has to be shown. Fig. 4D CTB binding is very different between HeLa and T47D. This has to be discussed. Is the membrane density of GM1 similar between HeLa and T47D?

Does crowding affect CTB binding to GM1? Is there a possibility that CTB selectively binds GM1 in less-crowded area?

Although the authors discuss MUC1, they did not experimentally examine the effect of MUC1 on antibody binding. The effect of expression level of MUC1 on antibody binding has to be examined.

Reviewer #3 (Remarks to the Author):

This paper presents an original approach to measure crowding of cell membranes in three dimensions. This is certainly innovative work, and merits publication in a journal with broad impact like Nature Communications in my view.

Overall the paper is well written and consists of a number of important steps using model systems to show the validity of the approach. The experimental results are furthermore backed up by theoretical considerations and coarse-grain simulations, providing a convincing story.

I only have a few remarks which could help in further improving the work:

1) When introducing the binding potential U , it might be good to explain to what extent this represents a free energy. Later in the manuscript the authors refer to it as a free energy, but it is introduced as a potential. Also the physical origin of the repulsive part of this potential (the crowding potential) could be explained - I assume it is mostly due to entropic factors ?

2) One of the questionable simplifications underlying the theoretical and modelling part of this work is the description of disordered proteins and glycans as ideal polymers. For PEG this might be reasonable, but for biopolymers in general this is likely not the case - in particular glycans are known to be able to form gel-like meshes that are quite distinct from ideal polymers. Some additional calculations showing how the choice of model parameters could impact the predictions would be beneficial in this regard (e.g., invoking some self-interactions between the polymer beads). At least the possible drawbacks of these simplifications should be discussed.

3) On page 10 the authors write "While the absolute magnitudes of observed ΔU were higher on beads than GUVs, we attribute this difference to the lower membrane friction on GUVs enabling IgG to more easily exclude PEG2k when binding." I do not follow this line of argument, 'more easily exclude' what does this mean ? Is this, or can this, be backed up with additional simulations or theory ?

4) Line 412: typo 'advance advance'

Response to Reviews

For the manuscript *Antibody binding reports heterogeneities in cell membrane organization* by D. Arnold, Y. Xu, and S. Takatori for *Nature Communications*

We thank the reviewers for their helpful comments and appreciate the time they devoted to improving our manuscript. We have considered their comments and suggestions carefully and have made extensive revisions to the manuscript. Importantly, we have conducted additional experiments to measure the height-dependence of crowding on red blood cells treated with neuraminidase to remove sialic acid from the glycocalyx. This experiment showed a significant reduction in crowding, confirming that glycophorin A is a major contributor to red blood cell surface crowding. In addition, we have modified our interpretation of the increased antibody binding to cholera toxin B. We discuss the possibility that cholera toxin oligomerization may exclude bystander proteins, leading to spatially heterogeneous crowding. We then confirm this possibility with an additional control experiment that uses annexin V, another surface-clustering protein, as an antigen target. We conclude by emphasizing that regardless of the mechanism, whether through clustering or lipid rafts, our work presents the first measurement of nanoscale crowding heterogeneities on live cells using molecular probes.

We have made extensive revisions to the main text and the Supplementary, and the modified sections are in **red**. In this document, reviewer comments are reproduced in black and our point-by-point responses are in **blue**.

Reviewer 1:

In this manuscript, Arnold, Xu, and Takatori apply their unique methodology for nanoscale measurements of distance from a membrane surface to build a series of probes for measuring the impact of crowding on antibody-antigen binding. The probes are first tested in synthetic systems, then a theoretical model and coarse-grained simulations are used to describe and validate the approach. The probes are then applied to the surface of red blood cells and tumor cell lines to test the impact of endogenous crowding molecules on antigen detection. One key conclusion is a difference in crowding between different membrane domains on the cell surface.

Overall, the manuscript is well written, well-reasoned, and adds important evidence to an accumulating body of research on the biophysical impacts of the cell's boundary layer on the activity of molecules at the cell surface. While I am supportive of publication, I believe the manuscript could be improved by the following considerations:

We thank the reviewer for the concise summary of our work, and agree with the characterization of the importance of pericellular biophysical interactions on cell surface activity.

1. While Fig1 uses a series of probes where the longest probe exceeds the height of the crowding molecule (10k = 7 nm, PEG2k crowder = ~4 nm), the same is not true for RBCs in Fig 3. A longer probe should be tested such that the probe height exceeds the expected GYPA height so that parity between crowded and non-crowded antibody binding can be demonstrated. Alternatively, cleavage or removal of GYPA could be used to show how modifying the crowding of RBCs impacts antibody binding

We conducted additional experiments in which we treated red blood cells with neuraminidase (NA) to cleave N-Acetylneuraminic acid (sialic acid) from surface glycoproteins and then remeasured crowding using our cholesterol-PEG-FITC sensors. The results are presented in Fig. 3 and demonstrate an overall reduction in crowding over that measured on wild type red blood cells, which flattens to an approximately constant value far from the surface. This demonstrates that the removal of negatively-charged sugars causes the cell surface glycocalyx to decrease in height, allowing our existing sensors to probe heights above the crowders.

We have added a paragraph discussing this additional experiment on Page 9 of the main text, in which we also provide context that Takatori and Son et al. (*bioRxiv*, 2022) showed that removing sialic acid with NA causes the red blood cell glycocalyx height to decrease. Through simulations, they showed that reducing the Debye screening length in a polymer brush subject to a Yukawa potential can cause a polymer brush to de-swell, suggesting that removing these charged species entirely may cause the same effect. However, in reviewing the thorough description of RBC surface glycosylation by Aoki (*Membranes*, 2017), we also note that approximately one-third to one-half of the sugars in the O-glycans on GYPA are sialic acid, and that there are 15 O-glycans on GYPA for every one N-glycan. Thus, thorough treatment with NA also leads to a significant reduction in the GYPA glycan size, which will also reduce its impact on crowding.

2. Cholera toxin is used as a raft-domain marker on cell lines to demonstrate that raft domains are less crowded than bulk membrane. However, glycolipid binding toxins can self-associate to form oligomers on the surface, including invaginations (see work by Johannes). Such cell surface domains may be quite

tightly packed with toxin and exclude other proteins independent of the underlying lipid composition. A more direct way to assay raft dependent behavior may be to measure binding to a monomeric raft marker such as a GPI-anchored protein. Put another way, data in Fig 4C, D supporting the claim that "raft-like membrane domains exclude bulky membrane proteins and glycoproteins" rely on the assumption that CTB is marking raft domains and is not producing its own organization. Mucl (which has been reported to be present in detergent resistant membranes) could still be present in raft domains, just not the same raft domains that underly CTB clusters.

We acknowledge the reviewer's comment that the domains to which cholera toxin B binds are not representative of lipid rafts as a whole, and that other effects like CTB oligomerization and clustering may deplete spectator protein density, creating a less-crowded microenvironment. Thus, we have modified our interpretation of these results to focus on the spatial heterogeneities in the composition of the glycocalyx specifically induced by CTB. This likely involves both changes in the lipid composition, which may in turn exclude transmembrane proteins (discussed in the next paragraph) but could also result from the simple clustering of CTB, as a new control experiment shows.

We have added additional text discussing the Hammond, et al. (*Proc. Nat. Acad. Sci. U.S.A.* 2005) reference listed in a comment below, with emphasis on their findings that bound cholera toxin triggered the condensation of 2D phase-separated domains on GUVs. We note that adding cholera toxin caused the transmembrane domain of LAT protein to exclude from the resultant CTB-rich phase. The latter point is important because LAT is primarily a transmembrane and cytoplasmic protein. The extracellular domain contains only four amino acids (the wild type contains two palmitoylations as well, but Hammond et al. expressed an un-palmitoylated version). Thus, this protein is almost certainly excluded via a shift in the microstructure and composition of the lipids, rather than being crowded out by CTB. Thus, this suggests that CTB can influence its surrounding protein composition by altering the lipid composition and packing. Thus it creates its own membrane raft-like domains, rather than acting as more of a passive marker of lipid raft location/presence. We therefore have changed our interpretation of our IgG/CTB binding results, as CTB domains are special and not necessarily representative of lipid rafts in general.

We have also added a brief discussion of Windschiegl et al. (*PLoS*, 2009). As the reviewer mentioned, Johannes' work here focuses on the effect of a ganglioside-binding toxin (in this case Shiga toxin) on the composition of reconstituted lipid domains. Of note, upon adding Shiga toxin to a phase-separated membrane, the authors find that that the distribution of perylene shifts from favoring the Lo phase to the Ld phase. This further suggests that the condensates formed by CTB not only feature an extracellular domain enriched in CTB, but also involve a redistribution of lipids. While not representative of all lipid rafts, this does suggest that our experiments are indeed probing laterally-heterogeneous membranes, and that the heterogeneity includes complexities mediated by lipids, as well as direct extracellular protein exclusion by CTB.

To address the possibility that CTB clustering may exclude crowding proteins, we have conducted an additional control experiment in which we use annexin V as our antigen probe on blebbed vesicles from HeLa cells. Annexin V binds to phosphatidylserine (PS) lipid headgroups and has been shown to associate into highly ordered clusters on the membrane surface (Andree et

al., *J. Biol. Chem.*, 1992; Lin, Chipot, and Scheuring, *Nat. Commun.*, 2020). Interestingly, Andree et al. showed that the clusters strongly prefer a planar orientation, to the point that they can turn spherical small unilamellar lipid vesicles polyhedral, with side lengths of ~100 nm. Moreover, Lin, Chipot, and Scheuring found annexin to be only about 2 nm tall, so like with CTB, we do not expect height to play a major role in modulating antibody binding.

The results of this annexin V control in Supplementary Fig. S12 indicate that antigen clustering may indeed be sufficient to exclude the glycocalyx from the local region, facilitating more favorable antibody binding. Because annexin typically only binds to apoptotic cells expressing PS on their surface, we were unable to get sufficient statistics on intact cells. Instead, we added annexin directly to cells in culture, and then measured antibody binding on blebs or vesicles that the cells had released, which bound annexin. We acknowledge that actin-bound proteins would not have made it into these GPMVs and blebs, but we believe that the similarity between relative $K_D/K_{D,0}$ here and in the CTB case suggests that protein oligomerization is likely contributing to the effect we observe with CTB vs. FITC binding, and that large protein oligomers are also likely to be less crowded than monomeric surface proteins. Ultimately, we have shown that these lateral heterogeneities exist on the cell membrane, and that antibody binding is an effective tool to probe these effects on nanometer length scales.

We have made extensive changes to Pages 10 -14 in the main text to address our comments above.

3. To make a stronger argument that Muc1 is the main crowding actor on tumor cell lines, analogous experiments to those in Fig 4D but without Muc1 should be performed. This could be accomplished by cell sorting using anti-Muc1 antibodies or PNA to separate high and low/no Muc1 cell populations as in Paszek et al. Another suggestion is to cleave cell surface mucin with a commercially available enzyme (Mucinase StcE, MilliporeSigma #SAE0202).

Upon further consideration, we believe that the relevant length scales to Muc1 crowding are inconsistent with the sizes of our molecular probes. We are focused on crowding heterogeneities on, or within about 10 nm of the cell surface, while mucins like Muc1 are hundreds of nanometers in size. Therefore, we still assert in the main text that HeLa and T47D are promising candidates for crowding studies, based primarily on the findings of Shurer et al. (*Cell*, 2019) that both wild type HeLa and T47D cells present surface tubes whose number densities are Muc1-dependent. This Muc1 dependence suggests that they experience considerable cell surface crowding. However, we have removed our claims connecting our findings regarding spatial crowding heterogeneities to Muc1 expression, as this connection cannot directly be drawn from the data we present, and Muc1 crowding may not be as relevant to such small probes (IgG is ~10 nm in size and Muc1 is ~100-1000 nm).

Minor recommendations/comments:

- In my opinion, the current title is not sufficiently informative. For the sake of future readers, I suggest a more specific, descriptive title

After careful consideration of both the current titles and alternative titles, we have concluded that our current title is the most descriptive. In this manuscript, we develop new techniques to

measure spatial heterogeneities in cell surface organization, with quantitative measurement of antibody binding being the common thread between our different measurements. We considered the alternative title scheme commonly employed by many authors, in which the title highlights a specific result or finding, but we felt it was more important to highlight the antibody/technique aspect of our work than a specific result regarding RBCs or mammalian cancer cells.

- The authors report a predicted GYPA height for the experiments in Fig 1C. Since GYPA is recombinant in these experiments, they could add a tag at the "top" and directly measure the height of GYPA.

We measured the height of an anti-GYPA antibody that binds to the N-terminus via cell surface optical profilometry (Son et al., *Proc. Nat. Acad. Sci. U.S.A.* 2021), finding it to be about 12 nm tall, which is consistent with our original approximation. This is now reflected in the main text discussion of Fig. 1C, as well as the Supplementary description of RBC glycocalyx coarse-graining for MD simulations.

- More discussion of the small discrepancy between theory and simulations in Fig 3B would be helpful.

We have added two sentences to the main text to better clarify the discrepancy, and in particular why there is a kink in the theory curve. The height-dependent monomer distribution given by Milner, Witten, and Cates theory is derived only for a monodisperse polymer brush. Here we add a second species, which we treat as a second polymer brush that does not interact with the first. Thus, we superimpose the polymer brush densities, which are exactly zero for all z greater than the brush height. This imposes a kink in our final profile of ΔU versus height.

- The authors assume that their chol-PEG-FITC probes are equally distributed in raft and non-raft domains at the plasma membrane based on partitioning experiments in GUVs. This assumption may be questioned because the lipid composition of GUVs does not fully represent PM composition, which may impact chol-PEG-FITC partitioning. GPMVs are a better model to report chol-PEG-FITC partitioning for raft-like PM domains and have been reported to show slight enrichment of chol-PEG 2k-FITC in raft domains (<https://doi.org/10.1101/2022.08.02.502487>).

We have cited this article alongside the reference to our GUV control showing approximately equal partitioning between Lo and Ld. Given that the enrichment listed in table S1 of the Levental preprint is only 26%, we will assume that any enrichment of chol-PEG0.5k-FITC in the raft domains is small compared to the raft partitioning of CTB.

Levental et al. also show that the identity of the molecule conjugated to the end of the PEG chain significantly affects phase partitioning, as cholesterol-PEG-biotin partitions much more strongly into Lo than cholesterol-PEG-FITC. Thus one may also surmise that if FITC is a Lo-phobic moiety, then the length of the PEG spacer may impact its partitioning as well, with a shorter spacer possibly shifting K_{eq} in favor of Ld. The Levental materials section does not list the molecular weight of PEG used in their cholesterol-PEG-FITC constructs, but if it is longer than 0.5 kDa, then this may also explain the discrepancy between our supplemental data and theirs.

We also bring up Windschiegel et al. (*PLoS*, 2009) again briefly, just to mention that they showed Shiga toxin excluding perylene from Lo domains on reconstituted bilayers. The different toxin, dye, and lipids add clear caveats to any comparison between mammalian cells and in-vitro experiments, but they do suggest an increased ordering in domains containing clustered gangliosides that may be especially hostile to impurities like cholesterol-PEG-dye.

We ultimately conclude that while it is difficult to say whether cholesterol-PEG-FITC will enrich in CTB clusters, we expect that chol-PEG-FITC enrichment will be much smaller than the local enrichment of CTB.

- Consider referencing Hammond et al PNAS 2005 which shows the coalescence of large membrane domains upon GM1 crosslinking by cholera toxin.

We have added text to the results section discussing the results from this paper, with more detail discussed in response to comment #2.

- Typo in reference 33 - Dan Fletcher listed three times.

The typo has been corrected.

- Line 79 - The Levental lab has reported the exclusion of bulky transmembrane domains from raft-like domains on GPMVs, but not whether there is an effect of cytoplasmic or extracellular domain size. However, Gurdap et al Biophys J 2022 has shown a decrease in ordered domain partitioning with increasing ectodomain mass/glycosylation, which would make a good reference here.

We have now cited this work in both the introduction and the results section and thank the reviewer for bringing this reference to our attention.

Reviewer 2:

Fig. 1. How much cholesterol-PEGx-FITC was incorporated to the beads? Is the beads washed after incubation with cholesterol-PEGx-FITC? Were all cholesterol-PEGx-FITC incorporated equally? How about the amount of free cholesterol-PEGx-FITC? How much cholesterol-PEGx-FITC was detached during incubation? How about the concentration dependence of DOPE-PEG2k on the antibody binding? Does GM1/CTB show similar results with cholesterol-PEGx-FITC/IgG?

Red blood cells were washed five times with PBS after incorporating the cholesterol-PEG-FITC sensors, so that high concentrations of soluble sensors did not quench the antibody. The methods and results sections of the main text have been updated to specify this step.

While the absolute magnitude of the surface density is difficult to precisely characterize, the cholesterol-PEG-FITC was incorporated into each sample equally, as reflected by <15% variation in the FITC fluorescence signal on the red blood cells. We have added this data in supplementary figure S8.

We found that approximately 20% of cholesterol sensors are lost one hour after incubation, which is the period in which the dissociation constant measurement is performed. We made every effort to make our measurements quickly within the 30-60 minute window after IgG equilibration to minimize the effect of cholesterol unbinding between bulk concentration datapoints on the binding isotherm.

We have added supplemental figure S7 to show the dependence of DOPE-PEG2k concentration on antibody binding. We chose to use 3% DOPE-PEG2k because it appeared to give an approximately 50% reduction in antibody binding from a bare surface at a given antibody concentration.

Fig. 3. Similarity between simulation and experiment does not exclude the possibility that GYPA and Band 3 are major contributors of the inhibition of antibody binding. Cells without GYPA and/or Band 3 or deficient glycosylation need to be checked.

We conducted additional experiments in which we treated red blood cells with neuraminidase (NA) to cleave N-Acetylneuraminic acid (sialic acid) from surface glycoproteins and then remeasured crowding using our cholesterol-PEG-FITC sensors. The results are presented in Fig. 3 and demonstrate an overall reduction in crowding over that measured on wild type red blood cells, which flattens to an approximately constant value far from the surface.

In the accompanying text, we note the description of RBC surface glycosylation by Aoki (*Membranes*, 2017) which says that one-third to one-half of the sugars in the O-glycans on GYPA are sialic acid, and that there are 15 O-glycans on GYPA for every one N-glycan. In addition, Aoki 2017 notes that GYPA constitutes 85% of periodic acid-Schiff stain-positive proteins on the red blood cell surface, meaning most mucin-like and heavily glycosylated proteins are glycophorin. Thus, the fact that we see such a dramatic reduction in crowding after removing a significant fraction of sialic acid confirms that glycophorin is surface crowding.

It is a bit trickier to make this argument for Band 3. Most of the long N-glycans in Band 3 are capped by sialic acid, but sialic acid comprises a much smaller fraction of Band 3's mass. The fact that the NA-treated glycocalyx crowding has less spatial dependence far from the surface suggests that surface proximal proteins may dominate crowding more than in WT cells. This would be consistent with shorter proteins like Band 3, which is not significantly perturbed by the treatment, playing a larger role in surface crowding. It is likely that other shorter proteins also play a role in this crowding, but we chose to truncate our model at only GYPA and Band 3, based upon both abundance and extracellular size.

Fig. 4. Fig. 4C Fluorescence image of T47D cells has to be shown. Fig. 4D CTB binding is very different between HeLa and T47D. This has to be discussed. Is the membrane density of GM1 similar between HeLa and T47D? Does crowding affect CTB binding to GM1? Is there a possibility that CTB selectively binds GM1 in less-crowded area? Although the authors discuss MUC1, they did not experimentally examine the effect of MUC1 on antibody binding. The effect of expression level of MUC1 on antibody binding has to be examined.

A fluorescence image of T47D was added to Fig. 4C. The anti-CTB IgG images corresponding to the HeLa and T47D images presented in the main text are in supplemental figure S10.

We were unable to find information in the literature regarding the relative degree of GM1 presentation between T47D and HeLa cells. Our measurements suggest that GM1 presentation is higher on HeLa cells than T47D (supplemental Fig. S10), based upon increased CTB binding. We do not expect crowding to play as much of a role on CTB binding to the cell surface as it does to IgG as CTB is much smaller than IgG, and the repulsive effect of the glycocalyx brush scales as r^3 , where r is the protein size. Moreover, despite there being less CTB bound to T47D than HeLa cells, we see greater anti-CTB IgG binding to T47D (Fig. S10), which suggests the crowding environment on T47D is the cause of this reduction in binding. We have added further text discussing this in the results section.

We have modified our discussion of lateral heterogeneity to focus more on spatial heterogeneities in the composition of the glycocalyx associated with CTB clusters, rather than on lipid rafts in general. CTB is known to oligomerize and associate with caveolae, which may make the microenvironments around CTB unrepresentative of lipid rafts as a whole. For example, CTB oligomerization and clustering may deplete spectator protein density directly, creating a less-crowded microenvironment independent of lipid behavior. We have added other text to the paper discussing why the less-crowded environment around CTB may involve both changes in the lipid composition, as well as clustering. However, we have conducted an additional control experiment with the surface-clustering protein annexin V, which suggests antigen oligomerization and clustering may be sufficient to account for the reduced local crowding. However it is important to emphasize that our technique still demonstrates considerable lateral heterogeneity in crowding on live cell surfaces, and that both effects: protein oligomerization and lipid rafts are ubiquitous on the mammalian cell plasma membrane.

Therefore, having redirected the focus of this section to revolve around heterogeneities in protein composition, which could be induced via the introduction of a protein that forms 2D membrane clusters, it should not matter whether CTB selectively binds to certain parts of the cell over

others, whether due to crowding, caveolae-induced invaginations, etc. At the nanoscale, the microenvironment around CTB reduces the local crowding experienced by the antibody, relative to the crowding spatially averaged over the entire cell.

Upon further consideration, we believe that the relevant length scales to Muc1 crowding are inconsistent with the sizes of our molecular probes. We are focused on crowding heterogeneities on, or within about 10 nm of the cell surface, while mucins like Muc1 are hundreds of nanometers in size. Therefore, we still assert in the main text that HeLa and T47D are promising candidates for crowding studies, based primarily on the findings of Shurer et al. (*Cell*, 2019) that both wild type HeLa and T47D cells present surface tubes whose number densities are Muc1-dependent. This Muc1 dependence suggests that they experience considerable cell surface crowding. However, we have removed our claims connecting our findings regarding spatial crowding heterogeneities to Muc1 expression, as this connection cannot directly be drawn from the data we present, and Muc1 crowding may not be as relevant to such small probes (IgG is ~10 nm in size and Muc1 is ~100-1000 nm).

Reviewer 3:

This paper presents an original approach to measure crowding of cell membranes in three dimensions. This is certainly innovative work, and merits publication in a journal with broad impact like Nature Communications in my view.

Overall the paper is well written and consists of a number of important steps using model systems to show the validity of the approach. The experimental results are furthermore backed up by theoretical considerations and coarse-grain simulations, providing a convincing story.

I only have a few remarks which could help in further improving the work:

We thank the reviewer for the concise summary of our manuscript and support for publication in *Nature Communications*. We appreciate the thoughtful comments and have incorporated them to craft a more theoretically robust and rigorous manuscript.

1) When introducing the binding potential U , it might be good to explain to what extent this represents a free energy. Later in the manuscript the authors refer to it as a free energy, but it is introduced as a potential. Also the physical origin of the repulsive part of this potential (the crowding potential) could be explained - I assume it is mostly due to entropic factors ?

We thank the reviewer for raising this point, and the reviewer is absolutely correct in that the binding potential U_{net} , defined before Eq. 1, represents a change in free energy from antibodies existing in bulk suspension to antibodies bound to the FITC antigen at a position z above the crowded surface. Likewise, the crowding potential ΔU reflects the free energy penalty from antibodies in bulk suspension inserting to a position z above the crowded surface. We have modified our usage of free energy versus interaction potential throughout the manuscript to reflect this important distinction.

The reviewer is also correct that the physical origin of the repulsive crowding potential ΔU is likely entropic: insertion of the antibody into the brush reduces the volume available to the polymer and thus reduces the number of polymer configurations. As Halperin (*Langmuir*, 1999) discusses, the increased osmotic pressure of the brush results in a repulsive interaction. We have added this explanation to our results section.

2) One of the questionable simplifications underlying the theoretical and modelling part of this work is the description of disordered proteins and glycans as ideal polymers. For PEG this might be reasonable, but for biopolymers in general this is likely not the case - in particular glycans are known to be able to form gel-like meshes that are quite distinct from ideal polymers. Some additional calculations showing how the choice of model parameters could impact the predictions would be beneficial in this regard (e.g., invoking some self-interactions between the polymer beads). At least the possible drawbacks of these simplifications should be discussed.

We thank the reviewer for bringing up this consideration. We acknowledge here and in the main text that our ideal polymer description deviates from realistic biopolymers in several ways. For

instance, GYPA contains negatively charged sialic acid groups which contribute to red blood cell surface crowding, as Takatori and Son et al. demonstrated (*bioRxiv*, 2022). Analogous to polyelectrolyte brushes, these charge interactions between polymer beads swell the biopolymer to more stretched configurations. To account for this effect, the fidelity of our Brownian dynamics simulations to an actual cell surface might be further enhanced by following the methods outlined in Takatori and Son et al., where polymer segments interact through both hard-sphere interactions as well as screened Coulomb potentials. Because the Debye length in the buffer is 0.7 nm, the addition of charges to the polymers play a minor quantitative role. At our level of coarse-graining, each “monomer” in the simulation is designed to model a grouping of many amino acids and the aggregated effects of charges are not straightforward to include. Our goal in this manuscript is to present a height-dependent crowding profile of a crowded surface, and we decided to proceed with a neutral brush at our level of coarse-graining to capture the experimental trends while reducing the number of adjustable parameters in our simulations. We have added a description in the main text in Page 7 to address the comments above.

We also thank the reviewer for bringing up the fact that glycans can adopt gel-like meshes, which are structurally distinct from polymer brushes. The permeability of gels is dictated by physical gel-particle interactions as well as the particle size relative to the mesh-size of the network. Lieleg et al. (*Biomacromolecules*, 2012) showed that particles as small as 50 nm can get trapped by a mucin gel. Although our IgG is smaller than 50 nm, we cannot conclude that it will penetrate freely into the red blood cell surface without knowledge of the glycan mesh-size. Moreover, Curnutt et al. (*Scientific Reports*, 2020) showed that mucin gelation occurs at relatively low pH (4-5), which is below that of our system, and of blood. We have added this discussion in Page 7 of the main text.

Another important simplification we make in simulations is the coarse-graining of biopolymers and IgG. As stated in the methods, the two side branches of each N-glycan on Band 3 were coarse-grained into a single chain. In doing so, we have neglected the entropic penalty of confining these side branches. Also, the IgG has a Y-shaped structure consisting of one fragment crystallizable (Fc) region and two fragment antigen-binding (Fab) region. One potential drawback to simulating the IgG as a spherical particle is the overestimation of the free energy penalty when inserted into the brush. Furthermore, only the Fab regions target and bind to the antigen, so our measured IgG affinity could be different from reality (Fig. 2a). However, because our main simulation result is the brush crowding potential ΔU , we do not expect the IgG coarse-graining to have significant consequences other than a small overestimation of the osmotic penalties associated with insertion. We have added this description in the Supplementary text, Page 7.

3) On page 10 the authors write "While the absolute magnitudes of observed ΔU were higher on beads than GUVs, we attribute this difference to the lower membrane friction on GUVs enabling IgG to more easily exclude PEG2k when binding." I do not follow this line of argument, 'more easily exclude' what does this mean? Is this, or can this, be backed up with additional simulations or theory?

Upon further consideration of this question, as well as additional experience conducting experiments involving GUVs, we have changed the main text in Page 11 to reflect a different

hypothesis for why ΔU is lower on GUVs than on beads. In this work, we assume that the final mole fractions of lipids in our reconstituted bilayers match the initial composition of lipid solution that we measure out from stocks dissolved in chloroform. Empirically, when forming GUVs through electroformation through the process described by Angelova and Dimitrov (*Faraday Discuss. Chem. Soc.*, 1986), we have found that the behavior of GUVs containing lipids with more hydrophilic headgroups, such as PEG and charged moieties like carboxylic acids and quaternary amines often suggests a low uptake of the more hydrophilic species. For example, Pramanik et al. (*Soft Matter*, 2022) found that nickel-NTA-conjugated lipid uptake in GUVs rises by a factor of 5 when GUVs are electroformed on platinum wires as opposed to indium tin oxide-coated glass (as we use in this paper). It is more likely that the antibodies bind more readily to GUVs simply because there is less PEG presented on the surface than on supported lipid bilayers on beads, as opposed to a more kinetic argument based upon the enhanced lateral fluidity of PEGylated lipids on GUVs versus beads.

4) Line 412: typo 'advance advance'

The typo has been corrected.

Summary of changes

The following is a list of major changes made, starting with the main text and then proceeding to the supplementary material. Changes are made in chronological order

- Page 3: changed references to Gurdap et al. from Levental et al., per reviewer 1's recommendation
- Page 5, 18, SI page 3: added text describing CSOP measurement of GYPA height
- Page 6, 9, 21: clarify that ΔU is a free energy rather than potential
- Page 7, SI page 6-7: we discuss potential challenges with and drawbacks of describing the glycocalyx as a polymer brush dominated by steric repulsions.
- Page 8, 9, 18 Fig. 3C: conducted an additional experiment using neuraminidase to remove RBC surface sialic acid and measured crowding as a function of height
- Page 8: added further discussion of assumptions made in RBC brush theory
- Page 9, 18: clarify that unbound sensors were washed from bulk before adding IgG
- Page 10-13: revised description of CTB-bound regions of the membrane to focus on the role of CTB in creating heterogeneity, rather than regarding CTB as a general marker of lipid rafts
- Page 13-14, 15 Fig. S12: conducted an additional control experiment on blebbed HeLa cells showing that annexin V is comparably crowded to CTB.
- SI page 15, Fig S7: justify choice of 3% PEG2k for reconstituted experiments by showing relative IgG binding in different PEG brushes.
- SI page 16, Fig. S8: shows consistent FITC sensor insertion amongst different PEG linkers.
- SI page 17, Fig. S9: shows wheat germ agglutinin binding to RBCs, confirming reduction in sialic acid upon neuraminidase treatment.
- SI page 18, Fig. S10: shows relative binding of CTB and anti-CTB IgG amongst the two cell lines.
- SI page 20, Fig. S12: shows annexin binding to HeLa blebbed vesicles, and compares annexin crowding to CTB and cholesterol-PEG-FITC.

REVIEWERS' COMMENTS

Reviewer #1 (Remarks to the Author):

The authors have done a commendable job with the revisions, which in my opinion address all the reviewer comments fully. The manuscript was interesting before and has now been amended to be more comprehensive in its conclusions. I am fully satisfied and supportive of publication.

Reviewer #2 (Remarks to the Author):

The authors adequately answered my concerns and now the paper is acceptable to the journal.

Reviewer #3 (Remarks to the Author):

The authors did a good job in addressing my comments, and those of the other referees; therefore I am happy to recommend the current manuscript for publication. No further changes are required.

Response to Reviews (second round)

For the manuscript *Antibody binding reports heterogeneities in cell membrane organization* by D. Arnold, Y. Xu, and S. Takatori for *Nature Communications*.

We thank the reviewers for their time in reviewing our paper a second time. We are pleased that all three were satisfied with our submission, and have not recommended any additional revisions before publication.

Reviewer #1:

The authors have done a commendable job with the revisions, which in my opinion address all the reviewer comments fully. The manuscript was interesting before and has now been amended to be more comprehensive in its conclusions. I am fully satisfied and supportive of publication.

We thank the reviewer for the detailed comments and recommendations in the first round. Their feedback enabled us to strengthen our arguments and we are pleased they have recommended publication.

Reviewer #2:

The authors adequately answered my concerns and now the paper is acceptable to the journal.

We thank the reviewer for taking the time to review our paper, and for providing constructive criticisms that ultimately strengthened our paper. We are grateful for their positive recommendation.

Reviewer #3:

The authors did a good job in addressing my comments, and those of the other referees; therefore I am happy to recommend the current manuscript for publication. No further changes are required.

We thank the reviewer for their rigorous review of our paper, and for enabling us to strengthen the thermodynamic arguments made in this work. We are pleased that they have recommended publication.